# Exponential Quantum Communication Advantage in Distributed Inference and Learning

**Dar Gilboa**[*][†]
Google Quantum AI
Venice, CA, United States

**Hagay Michaeli**[†]
Technion
Haifa, Israel

**Daniel Soudry**
Technion
Haifa, Israel

**Jarrod R. McClean**
Google Quantum AI
Venice, CA, United States

## Abstract

Training and inference with large machine learning models that far exceed the memory capacity of individual devices necessitates the design of distributed architectures, forcing one to contend with communication constraints. We present a framework for distributed computation over a quantum network in which data is encoded into specialized quantum states. We prove that for models within this framework, inference and training using gradient descent can be performed with exponentially less communication compared to their classical analogs, and with relatively modest overhead relative to standard gradient-based methods. We show that certain graph neural networks are particularly amenable to implementation within this framework, and moreover present empirical evidence that they perform well on standard benchmarks. To our knowledge, this is the first example of exponential quantum advantage for a generic class of machine learning problems that hold regardless of the data encoding cost. Moreover, we show that models in this class can encode highly nonlinear features of their inputs, and their expressivity increases exponentially with model depth. We also delineate the space of models for which exponential communication advantages hold by showing that they cannot hold for linear classification. Communication of quantum states that potentially limit the amount of information that can be extracted from them about the data and model parameters may also lead to improved privacy guarantees for distributed computation. Taken as a whole, these findings form a promising foundation for distributed machine learning over quantum networks.

## 1 Introduction

As the scale of the datasets and parameterized models used to perform computation over data continues to grow [62, 51], distributing workloads across multiple devices becomes essential for enabling progress. The choice of architecture for large-scale training and inference must not only make the best use of computational and memory resources, but also contend with the fact that communication may become a bottleneck [97]. This is particularly pertinent as models grow so large that they cannot rely on high-bandwidth interconnects within datacenters [17], but are instead trained across multiple datacenters [108]. When using modern optical interconnects, classical computers exchange bits represented by light. This however does not fully utilize the potential of the physical substrate; given suitable computational capabilities and algorithms, the *quantum* nature of light can be harnessed as a powerful communication resource. Here we show that for a broad class of parameterized models, if quantum bits (*qubits*) are communicated instead of classical bits, an exponential reduction in the communication required to perform inference and gradient-based training can be achieved. This protocol additionally guarantees improved privacy of both the user data and

---

[*]darg@google.com.
[†]Equal contribution.

38th Conference on Neural Information Processing Systems (NeurIPS 2024).

model parameters through natural features of quantum mechanics, without the need for additional cryptographic or privacy protocols. To our knowledge, this is the first example of generic, exponential quantum advantage on problems that occur naturally in the training and deployment of large machine learning models. These types of communication advantages help scope the future roles and interplay between quantum and classical communication for distributed machine learning.

Quantum computers promise dramatic speedups across a number of computational tasks, with perhaps the most prominent example being the ability revolutionize our understanding of nature by enabling the simulation of quantum systems, owing to the inherently quantum nature of many many physical phenomena [39, 72]. However, much of the data that one would like to compute with in practice seems to come from an emergent classical world rather than directly exhibiting quantum properties. While there are some well-known examples of exponential quantum speedups for classical problems, most famously factoring [106] and related hidden subgroup problems [31], these tend to be isolated and at times difficult to relate to practical applications that involve learning from data. In addition, even though significant speedups are known for certain ubiquitous problems in machine learning such as matrix inversion [48] and principal component analysis [73], the advantage is often lost when including the cost of loading classical data into the quantum computer or of reading out the result into classical memory. This is because the complexity of loading dense classical data into the amplitudes of a quantum state (which is typically the encoding needed to obtain an exponential runtime advantage) and of reading out the amplitudes from a quantum state into classical memory, are both polynomial in the number of amplitudes [1]. In applications where an efficient data access model avoids the above pitfalls, the complexity of quantum algorithms tends to depend on condition numbers of matrices which scale with system size in a way that reduces or even eliminates any quantum advantage [82]. It is worth noting that much of the discussion about the impact of quantum technology on machine learning has focused on computational advantage. However quantum resources are not only useful in reducing computational complexity — they can also provide an advantage in communication complexity, enabling exponential reductions in communication for some problems [101, 15]. Inspired by these results, we study a setting where quantum advantage in communication is possible across a wide class of machine learning models. This advantage holds without requiring any sparsity assumptions or elaborate data access models such as QRAM [42].

We focus on compositional distributed learning, known as *pipelining* [56, 16]. While there are a number of strategies for distributing machine learning workloads that are influenced by the requirements of different applications and hardware constraints [115, 61], splitting up a computational graph in a compositional fashion (Figure 1) is a common approach. We describe distributed, parameterized quantum circuits that can be used to perform inference over data when distributed in this way, and can be trained using gradient methods. The ideas we present can also be used to optimize models that use certain forms of data parallelism (Appendix C). In principle, such circuits could be implemented on quantum computers that are able to communicate quantum states.

We show the following:

- Even for simple distributed quantum circuits, there is an exponential quantum advantage in communication for the problem of estimating the loss and the gradients of the loss with respect to the parameters (Section 3). This additionally implies a privacy advantage from Holevo's bound (Appendix H). We also show that this is advantage is not a trivial consequence of the data encoding used, since it does not hold for certain problems like linear classification (Appendix E).

- We study a class of models that can efficiently approximate certain graph neural networks (Section 4), and show that they both maintain the exponential communication advantage and achieve performance comparable to standard classical models on common node and graph classification benchmarks (Section 5).

- For certain distributed circuits, there is an exponential advantage in communication for the entire training process, and not just for a single round of gradient estimation. This includes circuits for fine-tuning using pre-trained features. The proof is based on convergence rates for stochastic gradient descent under convexity assumptions (Appendix D).

- The ability to interleave multiple unitaries encoding nonlinear features of data enables expressivity to grow exponentially with depth, and universal function approximation in some settings. This implies that these models are highly expressive in contrast to popular belief about linear restrictions in quantum neural networks (Appendix F).

## 2 Preliminaries

### 2.1 Large-scale learning problems and distributed computation

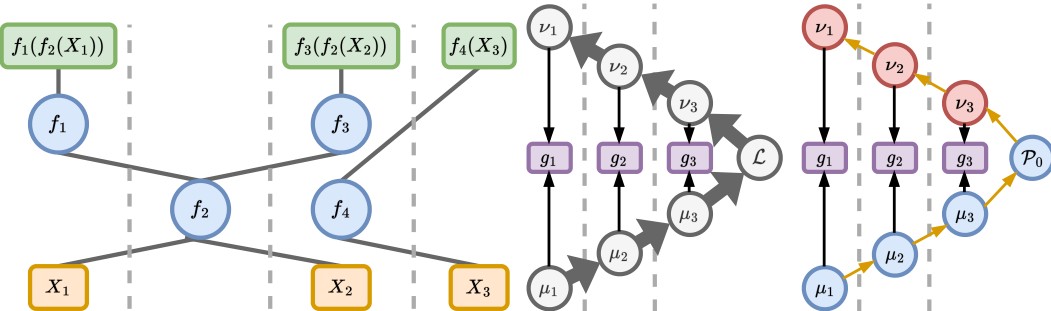

Figure 1: *Left:* Distributed, compositional computation. Dashed lines separate devices with computational and storage resources. The circular nodes represent parameterized functions that are allocated distinct hardware resources and are spatially separated, while the square nodes represent data (yellow) and outputs corresponding to different tasks (green). The vertical axis represents time. This framework of hardware allocation enables flexible modification of the model structure in a task-dependent fashion. *Right:* Computation of gradient estimators $g_\ell$ at different layers of a model distributed across multiple devices by pipelining. Computing forward features $\mu_\ell$ and backwards features $\nu_\ell$ (also known as computing a forward or backward pass) requires a large amount of classical communication (grey) but an exponentially smaller amount of quantum communication (yellow). $\mathcal{L}$ is the classical loss function, and $\mathcal{P}_0$ an operator whose expectation value with respect to a quantum model gives the analogous loss function in the quantum case.

Pipelining is a commonly used method of distributing a machine learning workload, in which different layers of a deep model are allocated distinct hardware resources [56, 87]. Training and inference then require communication of features between nodes. Pipelining enables flexible changes to the model architecture in a task-dependent manner, since subsets of a large model can be combined in an adaptive fashion to solve many downstream tasks. Additionally, pipelining allows sparse activation of a subset of a model required to solve a task, and facilitates better use of heterogeneous compute resources since it does not require storing identical copies of a large model. The potential for large models to be easily fine-tuned to solve multiple tasks is well-known [25, 20], and pipelined architectures which facilitate this are the norm in the latest generation of large language models [99, 16]. Data parallelism, in contrast, involves storing multiple copies of the model on different nodes, training each on a subsets of the data and exchanging information to synchronize parameter updates. In practice, different parallelization strategies are combined in order to exploit trade-offs between latency and throughput in a task-dependent fashion [115, 61, 97]. Distributed quantum models were considered recently in [94], but the potential for quantum advantage in communication in these settings was not discussed.

### 2.2 Communication complexity

Communication complexity [117, 65, 98] is the study of distributed computational problems using a cost model that focuses on the communication required between players rather than the time or computational complexity. The key object of study in this area is the tree induced by a communication protocol whose nodes enumerate all possible communication histories and whose leaves correspond to the outputs of the protocol. The product structure induced on the leaves of this tree as a function of the inputs allows one to bound the depth of the tree from below, which gives an unconditional lower bound on the communication complexity. The power of replacing classical bits of communication with qubits has been the subject of extensive study [30, 23, 27]. For certain problems such as Hidden Matching [15] and a variant of classification with deep linear models [101] an exponential quantum communication advantage holds, while for other canonical problems such as Disjointness only a polynomial advantage is possible [102]. Exponential advantage was also recently shown for the problem of sampling from a distribution defined by the solution to a linear regression problem

[83]. While there are many models of both quantum and classical communication, our results apply to *randomized* classical communication complexity, wherein the players are allowed to exchange random bits independent of their problem inputs, and are allowed to output an incorrect answer with some probability (bounded away from $1/2$ for a problem with binary output). It is also worth noting that communication advantages of the type we demonstrate can be naturally related to space advantages in streaming algorithms that may be of interest even in settings that do not involve distributed training [103].

At a glance, the development of networked quantum computers may seem much more challenging than the already herculean task of building a fault tolerant quantum computer. However, for some quantum network architectures, the existence of a long-lasting fault tolerant quantum memory as a quantum repeater, may be the enabling component that lifts low rate shared entanglement to a fully functional quantum network [86], and hence the timelines for small fault tolerant quantum computers and quantum networks may be more coincident than it might seem at first. As such, it is well motivated to consider potential communication advantages alongside computational advantages when talking about the applications of fault tolerant quantum computers. In Appendix G we briefly survey approaches to implementing quantum communication in practice, and the associated challenges.

In addition, while we largely restrict ourselves here to discussions of communication advantages, and most other studies focus on purely computational advantages, there may be interesting advantages at their intersection. For example, it is known that no quantum state built from a simple (or polynomial complexity) circuit can confer an exponential communication advantage, however states made from simple circuits can be made computationally difficult to distinguish [59]. Hence the use of quantum pre-computation [57] and communication may confer advantages even when traditional computational and communication cost models do not admit such advantages due to their restriction in scope.

## 3 Distributed learning with quantum resources

In this work we focus on parameterized models that are representative of the most common models used and studied today in quantum machine learning, sometimes referred to as quantum neural networks [79, 38, 28, 104]. We will use the standard Dirac notation of quantum mechanics throughout. A summary of relevant notation and the fundamentals of quantum mechanics is provided in Appendix A. We define a class models with parameters $\Theta$, taking an input $x$ which is a tensor of size $N$. The models take the following general form:

**Definition 3.1.** $\{A_\ell(\theta_\ell^A, x)\}, \{B_\ell(\theta_\ell^B, x)\}$ *for* $\ell \in \{1, \ldots, L\}$ *are each a set of unitary matrices of size* $N' \times N'$ *for some* $N'$ *such that* $\log N' = O(\log N)$ [3]. *The* $\theta_\ell^A, \theta_\ell^B$ *are vectors of* $P$ *parameters each. For every* $\ell, i$, *we assume that* $\frac{\partial A_\ell}{\partial \theta_{\ell i}^A}$ *is anti-hermitian and has two eigenvalues, and similarly for* $B_\ell$ [4].

*The model we consider is defined by*

$$|\varphi(\Theta, x)\rangle \equiv \left( \prod_{\ell=L}^{1} A_\ell(\theta_\ell^A, x) B_\ell(\theta_\ell^B, x) \right) |\psi(x)\rangle, \tag{3.1}$$

*where* $\psi(x)$ *is a fixed state of* $\log N'$ *qubits.*

*The loss function is given by*

$$\mathcal{L}(\Theta, x) \equiv \langle \varphi(\Theta, x) | \mathcal{P}_0 | \varphi(\Theta, x) \rangle, \tag{3.2}$$

*where* $\mathcal{P}_0$ *is a Pauli matrix that acts on the first qubit.*

In standard linear algebra notation, the output of the model is a unit norm $N'$-dimensional complex vector $\varphi_L$, defined recursively by

$$\varphi_0 = \psi(x), \quad \varphi_\ell = A_\ell(\theta_\ell^A, x) B_\ell(\theta_\ell^B, x) \varphi_{\ell-1}, \tag{3.3}$$

---

[3]We will consider some cases where $N' = N$, but will find it helpful at times to encode nonlinear features of $x$ in these unitaries, in which case we may have $N' > N$.

[4]The condition on the derivatives is in fact satisfied by many of the most common quantum neural network architectures [28, 34, 104]. It is satisfied for example if $A_\ell = \prod_{j=1}^{P} e^{i\alpha_{\ell j}^A \theta_{\ell j}^A \mathcal{P}_{\ell j}^A}$ and the $\mathcal{P}_{\ell j}^A$ are Pauli matrices, while $\alpha_{\ell j}^A$ are scalars. Such models are naturally amenable to implementation on quantum devices, and for $P = \tilde{O}(N^2)$ any unitary over $\log N'$ qubits can be written in this form [90].

where the entries of $\varphi_L$ are represented by the amplitudes of a quantum state. The loss takes the form $\mathcal{L}(\Theta, x) = (\varphi_L^*)^T \mathcal{P}_0 \varphi_L$ where $*$ indicates the entrywise complex conjugate, and this definition includes the standard $L^2$ loss as a special case.

Subsequently we omit the dependence on $x$ and $\Theta$ (or subsets of it) to lighten notation, and consider special cases where only subsets of the unitaries depend on $x$, or where the unitaries take a particular form and may not be parameterized. Denote by $\nabla_{A(B)} \mathcal{L}$ the entries of the gradient vector that correspond to the parameters of $\{A_\ell\}(\{B_\ell\})$.

In the special case where $x$ in a unit norm $N$-dimensional vector, a simple choice of $|\psi(x)\rangle$ is the amplitude encoding of $x$, given by

$$|\psi(x)\rangle = |x\rangle = \sum_{i=0}^{N-1} x_i |i\rangle. \tag{3.4}$$

However, despite its exponential compactness in representing the data, a naive implementation of the simplest choice is restricted to representing quadratic features of the data that can offer no substantial quantum advantage in a learning task [55], so the choice of data encoding is critical to the power of a model. The interesting parameter regime for classical data and models is one where $N, P$ are large, while $L$ is relatively modest. For general unitaries $P = O(N^2)$, which matches the scaling of the number of parameters in fully-connected networks. When the input tensor $x$ is a batch of datapoints, $N$ is equivalent to the product of batch size and input dimension.

The model in Definition 3.1 can be used to define distributed inference and learning problems by dividing the input $x$ and the parameterized unitaries between two players, Alice and Bob. We define their respective inputs as follows:

$$\begin{aligned} \text{Alice}: &\quad |\psi(x)\rangle, \{A_\ell\}, \\ \text{Bob}: &\quad \{B_\ell\}. \end{aligned} \tag{3.5}$$

The problems of interest require that Alice and Bob compute certain joint functions of their inputs. As a trivial base case, it is clear that in a communication cost model, all problems can be solved with communication cost at most the size of the inputs times the number of parties, by a protocol in which each party sends its inputs to all others. We will be interested in cases where one can do much better by taking advantage of quantum communication.

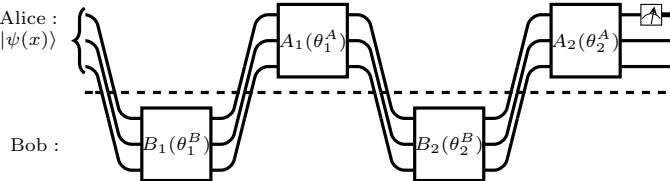

Figure 2: Distributed quantum circuit implementing $\mathcal{L}$ for $L = 2$. Both $\mathcal{L}$ and its gradients with respect to the parameters of the unitaries can be estimated with total communication that is polylogarithmic in the size of the input data $N$ and the number of trainable parameters per unitary $P$.

Given the inputs eq. (3.5), we will be interested chiefly in the two problems specified below.

**Problem 1** (Distributed Inference). *Alice and Bob each compute an estimate of $\langle \varphi | \mathcal{P}_0 | \varphi \rangle$ up to additive error $\varepsilon$.*

The straightforward algorithm for this problem, illustrated in fig. 2, requires $L$ rounds of communication. The other problem we consider is the following:

**Problem 2** (Distributed Gradient Estimation). *Alice computes an estimate of $\nabla_A \langle \varphi | \mathcal{P}_0 | \varphi \rangle$, while Bob computes an estimate of $\nabla_B \langle \varphi | Z_0 | \varphi \rangle$, up to additive error $\varepsilon$ in $L^\infty$.*

## 3.1 Communication complexity of inference and gradient estimation

We show that inference and gradient estimation are achievable with a logarithmic amount of quantum communication, which will represent an exponential improvement over the classical cost for some cases:

**Lemma 1.** *Problem 1 can be solved by communicating $O(\log N)$ qubits over $O(L/\varepsilon^2)$ rounds.*

Proof: Appendix B.

**Lemma 2.** *Problem 2 can be solved with probability greater than $1 - \delta$ by communicating $\tilde{O}(\log N (\log P)^2 \log(L/\delta)/\varepsilon^4)$ qubits over $O(L^2)$ rounds. The time and space complexity of the algorithm is $\sqrt{P} \, L \, \text{poly}(N, \log P, \varepsilon^{-1}, \log(1/\delta))$.*

Proof: Appendix B.

This upper bound is obtained by simply noting that the problem of gradient estimation at every layer can be reduced to a shadow tomography problem [7]:

**Theorem 1** (Shadow Tomography [3] solved with Threshold Search [13]). *For an unknown state $|\psi\rangle$ of $\log N$ qubits, given $K$ known two-outcome measurements $E_i$, there is an explicit algorithm that takes $|\psi\rangle^{\otimes k}$ as input, where $k = \tilde{O}(\log^2 K \log N \log(1/\delta)/\varepsilon^4)$, and produces estimates of $\langle\psi| E_i |\psi\rangle$ for all $i$ up to additive error $\varepsilon$ with probability greater than $1 - \delta$. $\tilde{O}$ hides subdominant polylog factors.*

Using reductions from known problems in communication complexity, we can show that the amount of classical communication required to solve this problem is polynomial in the size of the input, and additionally give a lower bound on the number of rounds of communication required by any quantum or classical algorithm:

**Lemma 3.**   *i) The classical randomized communication complexity of Problem 1 and Problem 2 with $\varepsilon < 1/2$ is $\Omega(\max(\sqrt{N}, L))$. [5]*

  *ii) Any algorithm (quantum or classical) for Problem 1 or Problem 2 requires either $\Omega(L)$ rounds of communication or $\Omega(N/L^4)$ qubits (or bits) of communication.*

Proof: Appendix B

The implication of the second result in Lemma 3 is that $\Omega(L)$ rounds of communication are necessary in order to obtain an exponential communication advantage for small $L$, since otherwise the number of qubits of communication required can scale linearly with $N$.

The combination of Lemma 1, Lemma 2 and Lemma 3 immediately implies exponential savings in communication for gradient estimation and inference:

**Theorem 2.** *If $L = O(\text{polylog}(N))$, $P = O(\text{poly}(N))$ and sufficiently large $N$, solving Problem 1 or Problem 2 with nontrivial success probability requires $\Omega(\sqrt{N})$ bits of classical communication, while $O(\text{polylog}(N, 1/\delta)\text{poly}(1/\varepsilon))$ qubits of communication suffice to solve these problems with probability at least $1 - \delta$.*

The regime where $L = O(\text{polylog}(N))$ is relevant for many classes of machine learning models. The required overhead in terms of time and space is only polynomial when compared to the straightforward classical algorithms for these problems.

The distribution of the model as in eq. (3.5) is an example of pipelining. Data parallelism is another common approach to distributed machine learning in which subsets of the data are distributed to identical copies of the model. In Appendix C we show that it can also be implemented using quantum circuits, which can then trained using gradient descent requiring quantum communication that is logarithmic in the number of parameters and input size.

Quantum advantage is possible in these problems because there is a bound on the complexity of the final output, whether it be correlated elements of the gradient up to some finite error or the low-dimensional output of a model. This might lead one to believe that whenever the output takes such a form, encoding the data in the amplitudes of a quantum state will trivially give an exponential advantage in communication complexity. We show however that the situation is slightly more nuanced, by considering the problem of inference with a linear model:

---

[5]The inputs to Problem 1 and Problem 2 are defined in terms of real numbers, which is seemingly incompatible with the setting of communication complexity which typically deals with finite inputs. However, similar (but slightly worse) lower bounds hold for discretized analogs of these problem that use $O(\log N)$ bits to represent the real numbers [101].

**Lemma 4.** *For the problem of distributed linear classification, there can be no exponential advantage in using quantum communication in place of classical communication.*

The precise statement and proof of this result are presented in Appendix E. This result also highlights that the worst case lower bounds such as Lemma 3 may not hold for circuits with certain low-dimensional or other simplifying structure.

# 4 Graph neural networks

The communication advantages in the previous section apply to relatively unstructured data and quantum circuits (essentially the only structure in the problem is related to the promise of the vector-in-subspace problem [101]), and it is a priori unclear how relevant they are to circuits that approximate useful neural networks, or act on structured data. Here we consider a class of shallow graph neural networks that achieve good performance on node classification tasks on large graphs [40]. We prove that an exponential quantum communication advantage still holds for this class of models.

Consider a graph with $N$ nodes. Define a local message passing operator on the graph $A$ which may be the normalized Laplacian or some other operator. Given some $N \times D_0$ matrix of graph features $X$, we consider models of the following form:

$$\Phi(X) = \mathcal{P}\left(\sigma(AXW_1)\right)W_2 \tag{4.1}$$

where $W_1 \in \mathbb{R}^{D_0 \times D_1}, W_2 \in \mathbb{R}^{D_1 \times D_2}$ are parameter matrices, $\sigma$ is a non-linearity and $\mathcal{P}$ is a sum pooling operator acting on the first index of its input, which can be represented as multiplication by an $N/t \times N$ matrix $\tilde{P}$. Since we would like a scalar output and a nonlinearity that can be implemented on a quantum computer, we instead compute

$$\varphi(X) = \text{tr}\left[\mathcal{P}\left(\sigma(AXW_1)\right)W_2\right], \tag{4.2}$$

with

$$\sigma(x) = ax^2 + bx + c \tag{4.3}$$

and $W_2 \in \mathbb{R}^{D_1 \times N/t}$ where $t$ is the size of the pooling window.

This allows us to define the following problem:

**Problem 3** (Quadratic graph network inference ($\text{QGNI}_{N,t}$))**.** *Alice is given $X$, Bob is given $A, W_1, W_2$. Only Alice is allowed to send messages. Their goal is to estimate $\varphi(X/\|X\|_F)$ to additive error $\varepsilon$.*

This models a scenario where only Bob has access to the connectivity of the graph, while Alice has access to the graph features. The normalization ensures that the choice of $X$ does not introduce a dependence of the final output on $N$.

In the following, we denote by $R_\varepsilon^\rightarrow$ and $Q_\varepsilon^\rightarrow$ the classical (public key randomized) communication complexity and quantum communication complexity respectively. We show:

**Lemma 5.** $R_\varepsilon^\rightarrow(\text{QGNI}_{N,t}) = \Omega(\sqrt{N/t})$ *for any* $\varepsilon \leq \frac{1}{4\left(t+\frac{1}{2}\right)\left(t+\frac{3}{2}\right)}$.

Proof: Appendix B

**Lemma 6.** $Q_\varepsilon^\rightarrow(\text{QGNI}_{N,t}) = O((|a|\,\alpha^2 + |b|\,\alpha)\left\|W_2\tilde{P}\right\|_\infty \log(ND_0)/\varepsilon)$ *where* $\alpha = \|W_1\|\,\|A\|$.

Proof: Appendix B

If this upper bound was a polynomial function of $N$, it would imply that an exponential communication advantage is impossible. For the parameter choices that realize classical communication lower bound, this is not the case, implying the following:

**Lemma 7.** *An exponential quantum advantage in communication holds for solving the inference problem* $\text{QGNI}_{N,t}$ *up to error* $\varepsilon \leq \frac{1}{4\left(t+\frac{1}{2}\right)\left(t+\frac{3}{2}\right)}$, *for any $t$ such that $t = \text{polylog}(N)$.*

Table 1: Test Accuracy for Node Classification and Decision Problem. Replacing PReLU with a polynomial of degree 2 causes a slight reduction in accuracy (less than 1%) for both node classification and decision problem across all datasets.

| | Node Classification | | | Decision Problem | | |
|---|---|---|---|---|---|---|
| Model | OGBN-Products | Reddit | Cora | OGBN-Products | Reddit | Cora |
| SIGN (PReLU) | $79.48 \pm 0.07$ | $96.55 \pm 0.02$ | $78.84 \pm 0.37$ | $84.39 \pm 1.73$ | $90.33 \pm 0.33$ | $88.10 \pm 5.61$ |
| SIGN (Poly) | $78.51 \pm 0.05$ | $96.31 \pm 0.03$ | $78.69 \pm 0.26$ | $83.70 \pm 1.48$ | $89.37 \pm 0.60$ | $87.14 \pm 3.92$ |

Proof: Appendix B. Note that this exponential advantage does not hold only for a single setting of the model weights, but rather for the entire family of models that can be used to solve $f - \mathrm{BHP}_{N,t}$ for functions $f$ that satisfy eq. (B.24).

Note that generically, one would not expect the numerator in the upper bound of Lemma 6 to scale polynomially with $N$. If $A$ is for example a normalized graph Laplacian then $||A|| \leq 2$. If we use a standard initialization scheme for the weights (say Gaussians with variance $1/(n_{in} + n_{out})$), the upper bound scales like $O((|a| + |b|) \log(ND_0)\mathrm{poly}(t, D_0, D_1)/\varepsilon)$ in expectation. Note that if the model output decays polynomially with $N$, the upper bound will not be useful since one would need to choose $\varepsilon$ to be inverse polynomial in $N$. This could happen for example in a classification task considered in Section 5.2.1 when the classes are exactly balanced, or when the network is untrained and not sensitive to the structure in the data. While it is difficult to argue analytically about the scaling out the network output or the norms of the weight matrices after training due to the nonlinearity of the dynamics, we empirically compute these and find that they remain controlled for the datasets we study (see Appendix J.3).

In Section 5, we show that models of the form studied here achieve good performance on standard benchmarks, commensurate with state of the art models. Of particular relevance are the graph classification problems considered in Section 5.2.1, where the output takes the form eq. (4.2).

## 5    Experimental results

### 5.1    Model

We evaluate our model[6] (as defined in Equation (4.1)) on several graph tasks using common benchmarks and the DGL library [113]. We use the SIGN model proposed by [40] as a baseline. The SIGN model can be seen as an instance of our model where the message passing operator $A$ represents a column stack of $R$ hops, the original features of $X$ are duplicated $R$ times and $W_1$ is a block diagonal matrix. In Section 5.2 we simply replace the PReLU activation function with a second-degree polynomial with trainable coefficients [80] and compare the models on three node classification tasks. In Section 5.3, we implement a more general form of SIGN by relaxing $W_1$ to be a dense matrix and evaluate our model over several graph-classification datasets.

### 5.2    Node classification

We evaluate our model on three public node classification datasets: OGBN-Products [54], Reddit [47], and Cora [78]. For both the baseline and polynomial model, we use SIGN with 5 hops of the neighbor averaging operator. We train on each dataset for 1000 epochs using Adam optimizer and report the test accuracy averaged on 10 runs (full details in Appendix J). Our results in Table 3 show that replacing the PReLU activation function with a second-degree polynomial causes a reduction of less than 1% on all of the tested datasets.

#### 5.2.1    Decision problems

We reduce the node classification task into a binary graph classification task by proposing the following decision problem: for a pair of classes $(c_1, c_2)$, return 1 if $c_1$ has more nodes; otherwise, return 0. We solve this task for each pair of classes by summing the node classification model output across all nodes and choosing the class with the higher score. We use the *node* classification training,

---

[6]Our code will be available at github.com/hmichaeli/quantum_gnns/.

Table 2: Graph Classification Test Accuracy. Our model achieves comparable results to GIN and other known models on most datasets (see full table in Table 5).

| Model | Dataset | | | | | | |
|---|---|---|---|---|---|---|---|
| | MUTAG | PTC | NCI1 | PROTEINS | COLLAB | IMDB-M | REDDIT-M |
| GIN [114] | 89.40±5.60 | 64.60±7.0 | 82.17±1.7 | 76.2 ±2.8 | 80.2 ±1.90 | 52.3 ±2.8 | 57.5±1.5 |
| DropGIN[92] | 90.4 ±7.0 | 66.3 ±8.6 | - | 76.3 ±6.1 | - | 51.4 ±2.8 | - |
| DGCNN[118] | 85.8 ±1.7 | 58.6 ±2.5 | - | 75.5 ±0.9 | - | 47.8 ±0.9 | - |
| U2GNN [89] | 89.97±3.65 | 69.63±3.60 | - | 78.53±4.07 | 77.84±1.48 | 53.60±3.53 | - |
| HGP-SL[119] | - | - | 78.45±0.77 | 84.91±1.62 | - | - | - |
| WKPI[120] | 88.30±2.6 | 68.10±2.4 | 87.5 ±0.5 | 78.5±0.4 | - | 49.5 ± 0.4 | 59.5 ± 0.6 |
| SIGN (ours) | 92.02±6.45 | 68.0 ±8.17 | 77.25±1.42 | 76.55±5.10 | 81.82±1.42 | 53.13±3.01 | 54.09±1.76 |

choose the model with the highest validation accuracy on the *graph* classification task, and report its accuracy on the test sets. The model output in this form is given by eq. (4.1).

## 5.3 Graph classification

We evaluate our model on several graph classification benchmarks: bioinformatics datasets (MUTAG, PTC, NCI1, PROTEINS) [105, 50, 35, 21] and social networks (COLLAB, IMDB-BINARY, IMDB-MULTI, REDDIT-BINARY, REDDIT-MULTI) [116]. For the bioinformatics datasets, we use the standard categorical node features. As proposed in [114], we use one-hot encoding of the node degree as node features for the COLLAB and IMDB datasets, and for REDDIT datasets all nodes are set with an identical scalar feature of 1. We convert the polynomial SIGN model in Section 5.2 into a graph classification model by inserting a SumPool operator as described in Equation (4.1). We use the sign diffusion operator [113] and stack $R_i$ instances of each of its four message passing operators, where $\{R_i\}_{i=1}^4$ are selected during a hyperparameter tuning, as well as the hidden dimension size and optimization setting (see Appendix J for more details). We follow the validation regime proposed by [114]; we perform 10-fold cross-validation, train each fold for 350 epochs using Adam optimizer, and report in Table 2 the maximal value and standard-deviation of the averaged validation accuracy curve. For all datasets, except for REDDIT, our model achieves comparable to or better than other commonly used models, despite most of them using multiple layers. While the results show that on most datasets our shallow architecture suffices given sufficient width in the message passing and hidden layer, we hypothesize that datasets without any node features (such as REDDIT) require at least two layers of message passing.

## 6 Discussion

This work constitutes a preliminary investigation into a generic class of quantum circuits that has the potential for enabling an exponential communication advantage in problems of classical data processing including training and inference with large parameterized models over large datasets, with inherent privacy advantages. Communication constraints may become even more relevant if such models are trained on data that is obtained by inherently distributed interaction with the physical world [37]. The ability to compute using data with privacy guarantees can be potentially applied to proprietary data. This could become highly desirable even in the near future as the rate of publicly-available data production appears to be outstripped by the growth rate of training sets of large language models [110].

A limitation of the current results is that it's unclear to what extent powerful neural networks can be approximated using quantum circuits, even though we provide positive evidence in the form of the results on graph networks in Section 4. Additionally, the advantages we study require deep $(\text{poly}(N))$, fault-tolerant quantum circuits. While this is a common feature of problems for which quantum communication advantages hold, the overhead of quantum error-correction in such circuits may be considerable. Detailed resource estimates would be necessary to understand better the practicality of this approach for achieving useful quantum advantage. Our results naturally raise further questions regarding the expressive power and trainability of these types of circuits, which may be of independent interest. We collect some of these in Appendix I.

## Acknowledgements

The authors would like to thank Amira Abbas, Ryan Babbush, Dave Bacon and Robbie King for helpful discussions and comments on the manuscript. The research of DS was Funded by the European Union (ERC, A-B-C-Deep, 101039436). Views and opinions expressed are however those of the author only and do not necessarily reflect those of the European Union or the European Research Council Executive Agency (ERCEA). Neither the European Union nor the granting authority can be held responsible for them. DS also acknowledges the support of the Schmidt Career Advancement Chair in AI.

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

# A   Notation and a very brief review of quantum mechanics

We denote by $\{a_i\}$ a set of elements indexed by $i$, with 1-based indexing unless otherwise specified, with the maximal value of $i$ explicitly specified when it is not clear from context. $[N]$ denotes the set $\{0, \ldots, N-1\}$. The complex conjugate of a number $c$ is denoted by $c^*$, and the conjugate transpose of a complex-valued matrix $A$ by $A^\dagger$.

We denote by $|\psi\rangle$ a vector of complex numbers $\{\psi_i\}$ representing the state of a quantum system when properly normalized, and by $\langle\psi|$ its dual (assuming it exists). The inner product between two such vectors of length $N$ is denoted by

$$\langle\psi|\varphi\rangle = \sum_{i=0}^{N-1} \psi_i^* \varphi_i. \tag{A.1}$$

Denoting by $|i\rangle$ for $i \in [N]$ a basis vector in an orthonormal basis with respect to the above inner product, we can also write

$$|\psi\rangle = \sum_{i=0}^{N-1} \psi_i |i\rangle. \tag{A.2}$$

Matrices will be denoted by capital letters, and when acting on quantum states will always be unitary. These can be specified in terms of their matrix elements using the Dirac notation defined above, as in

$$A = \sum_{ij} A_{ij} |i\rangle \langle j|. \tag{A.3}$$

Matrix-vector product are specified naturally in this notation by

Quantum mechanics is, in the simplest possible terms, a theory of probability based on conservation of the $L^2$ norm rather than the standard probability theory based on the $L^1$ norm [2, 90]. The state of a pure quantum system is described fully by a complex vector of $N$ numbers known as amplitudes which we denote by $\{\psi_i\}$ where $i \in \{0, \ldots, N-1\}$, and is written using Dirac notation as $|\psi\rangle$. The state is normalized so that

$$\langle\psi|\psi\rangle = \sum_{i=0}^{N-1} \psi_i^* \psi_i = \sum_{i=0}^{N-1} |\psi_i|^2 = 1, \tag{A.4}$$

which is the $L^2$ equivalent of the standard normalization condition of classical probability theory. It is a curious fact that the choice of $L^2$ requires the use of complex rather than real amplitudes, and that no consistent theory can be written in this way for any other $L^p$ norm [2]. The most general state of a quantum system is a probabilistic mixture of pure states, in the sense of the standard $L^1$-based rules of probability. We will not be concerned with these types of states, and so omit their description here, and subsequently whenever quantum states are discussed, the assumption is that they are pure.

Since any closed quantum system conserves probability, the $L^2$ norm of a quantum state is conserved during the evolution of a quantum state. Consequently, when representing and manipulating quantum states on a quantum computer, the fundamental operation is the application of a unitary matrix to a quantum state.

Given a quantum system with some discrete degrees of freedom, the number of amplitudes corresponds to the number of possible states of the system, and is thus exponential in the number of degrees of freedom. The simplest such degree of freedom is a binary one, called a qubit, which is analogous to a bit. Thus a state of $\log N$ qubits is described by $N$ complex amplitudes.

A fundamental property of quantum mechanics is that the amplitudes of a quantum state are not directly measurable. Given a Hermitian operator

$$\mathcal{O} = \sum_{i=0}^{N-1} \lambda_i |v_i\rangle \langle v_i| \tag{A.5}$$

with real eigenvalues $\{\lambda_i\}$, a measurement of $\mathcal{O}$ with respect to a state $|\psi\rangle$ gives the result $\lambda_i$ with probability $|\langle v_i|\psi\rangle|^2$. The real-valued quantity

$$\langle\psi|\mathcal{O}|\psi\rangle = \sum_{i=0}^{N-1} \lambda_i |\langle\psi|v_i\rangle|^2 \tag{A.6}$$

is the expectation value of $\mathcal{O}$ with respect to $|\psi\rangle$, and its value can be estimated by measurements. After a measurement with outcome $\lambda_i$, the original state is destroyed, collapsing to the state $|v_i\rangle$. A consequence of the fundamentally destructive nature of quantum measurement is that simply encoding information in the amplitudes of a quantum state dues not necessarily render it useful for downstream computation. It also implies that operations using amplitude-encoded data such as evaluating a simple loss function incur measurement error, unlike their classical counterparts that are typically limited only by machine precision. The design of quantum algorithms essentially amounts to a careful and intricate design of amplitude manipulations and measurements in order to extract useful information from the amplitudes of a quantum state. For a more complete treatment of these topics see [90].

## B  Proofs

*Proof of Lemma 1.* $\langle\varphi|\,\mathcal{P}_0\,|\varphi\rangle$ can be estimated by preparing $|\varphi\rangle$ and measuring it $O(1/\varepsilon^2)$ times. Preparing each copy of $|\varphi\rangle$ requires $O(L)$ rounds of communication, with each round involving the communication of a $\log N'$-qubit quantum state. Alice first prepares $|\psi(x)\rangle$, and this state is passed back and forth with each player applying $A_\ell$ or $B_\ell$ respectively for $\ell \in \{1, \ldots, L\}$.

$\square$

*Proof of Lemma 2.* We consider the parameters of the unitaries that Alice possesses first, and an identical argument follows for the parameters of Bob's unitaries.

We have

$$\frac{\partial}{\partial\theta_{\ell i}^A}\langle\varphi|\mathcal{P}_0|\varphi\rangle \quad =2\mathrm{Re}\,\langle\varphi|\,\mathcal{P}_0\prod_{k=L}^{\ell+1}A_k B_k \frac{\partial A_\ell}{\partial\theta_{\ell i}^A}B_\ell\prod_{k=\ell-1}^{1}A_k B_k\,|\psi(x)\rangle \tag{B.1}$$
$$\equiv 2\mathrm{Re}\,\langle\nu_{\ell i}^A|\mu_\ell^A\rangle$$

where

$$\left|\mu_\ell^A\right\rangle \quad =B_\ell\prod_{k=\ell-1}^{1}A_k B_k\,|\psi(x)\rangle\,,\quad \left|\nu_{\ell i}^A\right\rangle \quad =\left(\frac{\partial A_\ell}{\partial\theta_{\ell i}^A}\right)^{\dagger}\prod_{k=L}^{\ell+1}B_k^{\dagger}A_k^{\dagger}\mathcal{P}_0\,|\varphi\rangle\,, \tag{B.2}$$

correspond to forward and backward features for the $i$-the parameter of $A_\ell$ respectively. This is illustrated graphically in Figure 1. We also write

$$\left|\nu_{\ell 0}^A\right\rangle = \prod_{k=L}^{\ell+1}B_k^{\dagger}A_k^{\dagger}\mathcal{P}_0\,|\varphi\rangle\,. \tag{B.3}$$

Attaching an ancilla qubit denoted by $a$ to the feature states defined above, we define

$$\left|\psi_{\ell i}^A\right\rangle \equiv \frac{1}{\sqrt{2}}\left(|0\rangle\left|\mu_\ell^A\right\rangle + |1\rangle\left|\nu_{\ell i}^A\right\rangle\right), \tag{B.4}$$

and a Hermitian measurement operator

$$E_{\ell i}^A \equiv \quad \left(|0\rangle\langle 0|\otimes I + |1\rangle\langle 1|\otimes\left(\frac{\partial A_\ell}{\partial\theta_{\ell i}^A}\right)\right)X_a\left(|0\rangle\langle 0|\otimes I + |1\rangle\langle 1|\otimes\left(\frac{\partial A_\ell}{\partial\theta_{\ell i}^A}\right)^{\dagger}\right)$$
$$= \quad |1\rangle\langle 0|\otimes\left(\frac{\partial A_\ell}{\partial\theta_{\ell i}^A}\right) + |0\rangle\langle 1|\otimes\left(\frac{\partial A_\ell}{\partial\theta_{\ell i}^A}\right)^{\dagger}, \tag{B.5}$$

we then have

$$\langle\psi_{\ell 0}^A|\,E_{\ell i}^A\,|\psi_{\ell 0}^A\rangle \quad =\langle\psi_{\ell i}^A|\,X_a\,|\psi_{\ell i}^A\rangle$$
$$=\frac{\partial}{\partial\theta_{\ell i}^A}\langle\varphi|\,\mathcal{P}_0\,|\varphi\rangle\,, \tag{B.6}$$

where $X_a$ acts on the ancilla.

Note that $\left|\psi_{\ell 0}^A\right\rangle^{\otimes k}$ can be prepared by Alice first preparing $(|+\rangle|\psi(x)\rangle)^{\otimes k}$ and sending this state back and forth at most $2L$ times, with each player applying the appropriate unitaries conditioned on the value of the ancilla. Additionally, for any choice of $\ell$ and any $i$, Alice has full knowledge of the $E_{\ell i}^A$. They can thus be applied to quantum states and classical hypothesis states without requiring any communication.

The gradient can then be estimated using shadow tomography (Theorem 1). Specifically, for each $\ell$, Alice prepares $\tilde{O}(\log^2 P \log N \log(L/\delta)/\varepsilon^4)$ copies of $\left|\psi_0^{A_\ell}\right\rangle$, which requires $O(L)$ rounds of communication, each of $\tilde{O}(\log^2 P \log^2 N \log(L/\delta)/\varepsilon^4)$ qubits. She then runs shadow tomography to estimate $\nabla_{A_\ell} \langle\varphi|Z_0|\varphi\rangle$ up to error $\varepsilon$ with no additional communication. Bob does the same to estimate $\nabla_{B_\ell} \langle\varphi|Z_0|\varphi\rangle$. In total $O(L^2)$ rounds are needed to estimate the full gradient. The success probability of all $L$ applications of shadow tomography is at least $1 - \delta$ by a union bound.

Based on the results of [22], the space and time complexity of each application of shadow tomography is $\sqrt{P}\text{poly}(N, \log P, \varepsilon^{-1}, \log(1/\delta))$. This is the query complexity of the algorithm to oracles that implement the measurement operators $\left\{E_{\ell i}^Q\right\}$. Instantiating these oracles will incur a cost of at most $O(N^2)$. In cases where these operators have low rank the query complexity complexity will depend polynomially only on the rank instead of on $N$.

$\square$

*Proof of Lemma 3.* We first prove an $\Omega(\sqrt{N})$ lower bound on the amount of classical communication. Consider the following problem:

**Problem 4** ([101])**.** *Alice is given a vector $x \in S^{N-1}$ and two orthogonal linear subspaces of $\mathbb{R}^N$ each of dimension $N/2$, denoted $M_1, M_2$. Bob is given an orthogonal matrix $O$. Under the promise that either $\|M_1 Ox\|_2 \geq \sqrt{1-\theta^2}$ or $\|M_2 Ox\|_2 \geq \sqrt{1-\theta^2}$ for $0 < \theta < 1/\sqrt{2}$, Alice and Bob must determine which of the two cases holds.*

Ref. [101] showed that the randomized[7] classical communication complexity of the problem is $\Omega(\sqrt{N})$.

The reduction from Problem 4 to Problem 1 is obtained by choosing $\theta = 1/2$ and simply setting $L = 1, B_1 = O, |\psi(x)\rangle = |x\rangle, \mathcal{P}_0 = Z_0$, and

$$A_1 = \sum_{j=0}^{N/2-1} |0\rangle|j\rangle\langle v_j^1| + \sum_{j=0}^{N/2-1} |1\rangle|j\rangle\langle v_j^2|, \tag{B.7}$$

where the first register contains a single qubit and $\{v_j^k\}$ form an orthonormal basis of $M_k$, and picking any $\varepsilon < 1/2$. Note that this choice of $|\psi(x)\rangle$ implies $N' = N$. Estimating $\mathcal{L}$ to this accuracy now solves the desired problem since $\mathcal{L} = \langle x| O^T (\Pi_1 - \Pi_2) O |x\rangle$ where $\Pi_k$ is a projector onto $M_k$, and hence estimating this quantity up to error $1/2$ allows Alice and Bob to determine which subspace has large overlap with $Ox$.

The reduction from Problem 4 to Problem 2 is obtained by setting $L = 2$, picking $|\psi(x)\rangle, A_1, B_1$ as before, and additionally $B_2 = I, A_2 = e^{-i\theta_{2,1}^A X_0/2}$ initialized at $\theta_{2,1}^A = -\pi/2$. By the parameter shift rule [34], we have that if $U = e^{-i\theta\mathcal{P}/2}$ for some Pauli matrix $\mathcal{P}$, and $U$ is part of the parameterized circuit that defines $|\varphi\rangle$, then

$$\frac{\partial\mathcal{L}}{\partial\theta} = \frac{1}{2}(\mathcal{L}(\theta + \frac{\pi}{2}) - \mathcal{L}(\theta - \frac{\pi}{2})). \tag{B.8}$$

---

[7]In this setting Alice and Bob can share an arbitrary number of random bits that are independent of their inputs.

It follows that

$$
\begin{aligned}
\frac{\partial \mathcal{L}}{\partial \theta_{2,1}^A}\bigg|_{\theta_{2,1}^A = -\pi/2} &= \frac{1}{2}\left(\mathcal{L}(0) - \mathcal{L}(-\pi)\right) \\
&= \frac{1}{2}\left(\mathcal{L}(0) - \langle x| B_1^\dagger A_1^\dagger e^{-i\frac{\pi}{2}X_0} Z_0 e^{i\frac{\pi}{2}X_0} A_1 B_1 |x\rangle\right) \\
&= \frac{1}{2}\left(\mathcal{L}(0) - \langle x| B_1^\dagger A_1^\dagger X_0 Z_0 X_0 A_1 B_1 |x\rangle\right) \\
&= \frac{1}{2}\left(\mathcal{L}(0) + \langle x| B_1^\dagger A_1^\dagger Z_0 A_1 B_1 |x\rangle\right) \\
&= \mathcal{L}(0).
\end{aligned}
\tag{B.9}
$$

Estimating $\nabla_A \langle\varphi| Z_0 |\varphi\rangle$ to accuracy $\varepsilon < 2$ allows one to determine the sign of $\mathcal{L}(0)$, which as before gives the solution to Problem 4.

Next, we show that $\Omega(L)$ rounds are necessary in both the quantum and classical setting by a reduction from the bit version of pointer-chasing, as studied in [58, 96].

**Problem 5** (Pointer-chasing, bit version). *Alice receives a function $f_A : [N] \to [N]$ and Bob receives a function $f_B : [N] \to [N]$. Alice is also given a starting point $x \in [N]$, and both receive an integer $L_0$. Their goal is to compute the least significant bit of $f^{(L_0)}(x)$, where $f^{(1)}(x) = f_B(x)$, $f^{(2)}(x) = f_A(f_B(x)),\ldots$.*

Ref. [58] show that the quantum communication complexity of $L_0$-round bit pointer-chasing when Bob speaks first is $\Omega(N/L_0^4)$ (which holds for classical communication as well). This also bounds the $(L_0 - 1)$-round complexity when Alice speaks first (since such a protocol is strictly less powerful given that there are fewer rounds of communication). On the other hand, there is a trivial $L_0$-round protocol when Alice speaks first that requires $\log N$ bits of communication per round, in which Alice sends Bob $x$, he sends back $f^{(1)}(x)$, she replies with $f^{(2)}(f^{(1)}(x))$, and so forth. This, combined with the lower bound, implies as exponential separation in communication complexity as a function of the number of rounds.

To reduce this problem to Problem 1, we assume $f_A, f_B$ are invertible. This should not make the problem any easier since it implies that $f_A, f_B$ have the largest possible image. In this setting, $f_A, f_B$ can be described by unitary permutation matrices:

$$
U_A = \sum_i |f_A(i)\rangle \langle i|, \, U_B = \sum_i |f_B(i)\rangle \langle i|.
\tag{B.10}
$$

The corresponding circuit eq. (3.5) is then given by

$$
|\varphi\rangle = \text{SWAP}_{0 \leftrightarrow \log N - 1} U_B \ldots U_A U_B |x\rangle
\tag{B.11}
$$

in the case where Bob applies the function last, with an analogous circuit in the converse situation (if Bob performed the swap, Alice applies an additional identity map). Estimating $Z_0$ to accuracy $\varepsilon < 1$ using this state will then reveal the least significant bit of $f^{(L_0)}(x)$. This gives a circuit with $L$ layers, where $L_0 \le 2L - 1$. Thus any protocol with less than $L_0$ rounds (meaning less than $2L - 1$ rounds) would require communicating $\Omega(N/L_0^4) = \Omega(N/L^4)$ qubits, since the converse will contradict the results of [58]. The reduction to Problem 2 is along similar lines to the one described by eq. (B.9), with the state in that circuit replaced by eq. (B.11). This requires at most two additional rounds of communication.

Since quantum communication is at least as powerful than classical communication, these bounds also hold for classical communication. Since each round involves communicating at least a single bit, this gives an $\Omega(L)$ bound on the classical communication complexity. $\qquad\square$

*Proof of Lemma 5.* The proof is based on a reduction from the $f$-Boolean Hidden Partition problem ($f - \text{BHP}_{N,t}$) studied in [36]. This is defined as follows:

**Problem 6** (Boolean Hidden Partition [36] ($f - \text{BHP}_{N,t}$)). *Assume $t$ divides $N$. Alice is given $x \in \{-1,1\}^N$. Bob is given a permutation $\Pi$ over $[N]$, a boolean function $f : \{-1,1\}^t \to \{-1,1\}$, and a vector $v \in \{-1,1\}^{N/t}$. We are guaranteed that for any $k \in \{1,\ldots,N/t\}$,*

$$
f([\Pi x]_{[(k-1)t+1:kt]}) * v_k = s
\tag{B.12}
$$

*for some $s \in \{-1, 1\}$. Their goal is to determine the value of $s$.*

A polynomial $p_f : \{-1, 1\}^t \to \mathbb{R}$ is said to sign-represent a boolean function $f$ if $\text{sign}(p_f(y)) = f(y)$ for all $y \in \{0, 1\}^t$. The *sign-degree* of $f$ ($\text{sdeg}(f)$) is the minimal degree of a polynomial that sign-represents it. In the special case $\text{sdeg}(f) = 2$, $f - \text{BHP}_{N,t}$ can be solved with exponential quantum communication advantage [36]. For a vector $y \in \{0, 1\}^t$, define $\tilde{y} = (1, y_1, \ldots, y_t)$. It is also known that if $\text{sdeg}(f) = 2$, then there exists a sign-representing polynomial $p_f$ that can be written as

$$p_f(y) = \tilde{y}^T R \tilde{y} \tag{B.13}$$

for some matrix real $R$ [4]. Moreover, for any $f$ there exists such a $p_f$ with $\max_{x \in \{-1,1\}^t} |p_f(x)| \leq 3$. We denote by $\beta = \min_{x \in \{-1,1\}^t} |p_f(x)|$ the *bias* of $p_f$.

We now describe a reduction from $f - \text{BHP}_{N,t}$ with $\text{sdeg}(f) = 2$ to $\text{QGNI}_{CN,t}$ for some constant $1 \leq C \leq 3/2$. As is typical in communication complexity, the parties are allowed to exchange bits that are independent of the problem input, and these are not counted when measuring the communication complexity of a protocol that depends on the inputs. Before receiving their inputs, Alice thus sends two orthogonal vectors $u_0, u_1$ of length $D_0$ to Bob, with each entry described by $K$ bits [8].

Assume Alice and Bob are given an instance of $\text{BHP}_{N,t}$. They use it to construct an instance of $\text{QGNI}_{(t+1)N/t,t}$ with $D_1 = 1$. Alice constructs $X \in \mathbb{R}^{(t+1)N/t}$ by picking the rows $X_i$ according to

$$X_i = \begin{cases} \frac{1}{\sqrt{(t+1)N/t}} \left( \frac{1-x_i}{2} u_0^T + \frac{1+x_i}{2} u_1^T \right) & i \leq N \\ \frac{1}{\sqrt{(t+1)N/t}} u_1 & i > N \end{cases}. \tag{B.14}$$

Note that with this definition $||X||_F = 1$. Bob defines a permutation $\pi'$ over $[(t+1)N/t]$ by

$$\pi'(i) = \begin{cases} \lfloor i/t \rfloor (t+1) + i\%t + 1 & i \leq N \\ (i - N - 1)(t+1) + 1 & i > N \end{cases}, \tag{B.15}$$

denoting the corresponding permutation matrix $\Pi'$. Define by $\overline{x}$ the concatenation of Alice's input $x$ with $1^{(t+1)N/t}$. The purpose of this permutation is that $\Pi'\Pi\overline{x} \equiv \tilde{x}$ will be a concatenation of $N/t$ vectors of length $t+1$, with the $i$-th vector equal to $(1, [\Pi x]_{t(i-1)+1}, \ldots, [\Pi x]_{ti}) \equiv \tilde{x}_{(i)}$.

Note that we can assume wlog that $R$ in eq. (B.13) is symmetric since $p_f$ is independent of its anti-symmetric part. It can thus be diagonalized by an orthogonal matrix $U$, and denoting the diagonal matrix of its real eigenvalues by $D$, we define a (complex-valued) matrix $S = U\sqrt{D}$ that satisfies $R = SS^T$. Bob therefore defines his model by

$$A = (I_{N/t} \otimes S^T)\Pi'\Pi, \quad W_1 = u_1 - u_0, \quad W_2 = v^T. \tag{B.16}$$

Additionally, he picks the pooling operator $\mathcal{P} : \mathbb{R}^{(t+1)N/t} \to \mathbb{R}^{N/t}$ to be sum pooling with window size $t+1$ (i.e. $\mathcal{P}(x)_j = \sum_{k=(j-1)(t+1)+1}^{j(t+1)} x_k$). Bob also uses a simple quadratic nonlinearity by choosing $a = 1, b = c = 0$ in eq. (4.3).

---

[8]Since $D_0$ is arbitrary and in particular independent of $N$, even if we count this communication it will not affect the scaling with $N$ which the main property we are interested in. This independence is also natural since it implies that the number of local graph features is independent of the size of the graph.

To see that solving $\text{QGNI}_{(t+1)N/t,t}$ to error $\varepsilon < 1/2$ indeed provides a solution to $\text{BHP}_{N,t}$, note that

$$\mathcal{P}\left(\sigma(AXW_1)\right)_i = \mathcal{P}\left(\sigma(\frac{1}{\sqrt{(t+1)N/t}}A\bar{x})\right)_i \tag{B.17}$$

$$= \mathcal{P}\left(\sigma(\frac{1}{\sqrt{(t+1)N/t}}(I_{N/t}\otimes S^T)\Pi'\Pi\bar{x})\right)_i \tag{B.18}$$

$$= \mathcal{P}\left(\sigma(\frac{1}{\sqrt{(t+1)N/t}}(I_{N/t}\otimes S^T)\tilde{x})\right)_i \tag{B.19}$$

$$= \frac{1}{(t+1)N/t}\sum_{j=1}^{t+1}([S^T\tilde{x}_{(i)}]_j)^2 \tag{B.20}$$

$$= \frac{1}{(t+1)N/t}\tilde{x}_{(i)}^T SS^T\tilde{x}_{(i)} \tag{B.21}$$

$$= \frac{1}{(t+1)N/t}p_f([\Pi x]_{[(i-1)t+1:it]}). \tag{B.22}$$

Given the choice of $W_2$, one obtains

$$\varphi(X/||X||_F) = \frac{1}{(t+1)N/t}\sum_{i=1}^{N/t}p_f([\Pi x]_{[(i-1)t+1:it]})v_i. \tag{B.23}$$

It follows that $\text{sign}(\varphi(X)) = s$ and $|\varphi(X)| \geq \beta$. It is thus possible to decide the value of $s$ if $\varphi(X/||X||_F)$ is estimated to some error smaller than $\beta$.

From Theorem 4 of [36], we have $R^\rightarrow(f - \text{BHP}_{N,t}) = \Omega(\sqrt{N/t})$ for any $f$ that has sign-degree 2 and satisfies some additional conditions. The reduction then implies

$$R_\beta^\rightarrow(\text{QGNI}_{N,t}) = \Omega(\sqrt{(t/(t+1))N/t}). \tag{B.24}$$

This can be simplified by noting that since $t \geq 2$, $t/(t+1) \geq 2/3$. The lower bound in [36] is based on choosing $f$ which belongs to a specific class of symmetric boolean functions (meaning $f(y) = \tilde{f}(|y|)$ where $|y| = |\{i : y_i = -1\}|$). Specifically, $\tilde{f}$ is defined by the choice of $t$ and two additional integer parameters $\theta_1, \theta_2$ such that $0 \leq \theta_1 < \theta_2 < t$ and

$$\tilde{f}(|y|) = \begin{cases} 1 & 0 \leq |y| \leq \theta_1 \text{ or } \theta_2 < |y|, \\ -1 & \theta_1 < |y| \leq \theta_2, \end{cases} \tag{B.25}$$

(and an additional technical condition that will not be of relevance to our analysis).

We next construct a sign-representing polynomial $p_f$ for any $f$ that takes the above form, and compute its bias $\beta$. Since $f$ is symmetric of sign degree 2, it suffices to construct a polynomial $\tilde{p}_f : \mathbb{R} \to \mathbb{R}$ such that $p_f(y) = \tilde{p}_f(|y|)$ for this purpose. If we can produce some $\beta'$ that bounds $\beta$ from below for any choice of $t, \theta_1, \theta_2$, then the lower bound from Theorem 4 of [36] holds for any error smaller than $\beta'$.

We choose $\tilde{p}_f(z) = \tilde{a}z^2 + \tilde{b}z + 1$, with the constraints $\tilde{p}_f(\theta_1 + 1/2) = 0, \tilde{p}_f(\theta_2 + 1/2) = 0$. These lead to the solution

$$\tilde{p}_f(z) = \frac{1}{\theta_1^+\theta_2^+}z^2 - \frac{1}{\theta_1^+\theta_2^+}\frac{\theta_2^{+2} - \theta_1^{+2}}{\theta_2^+ - \theta_1^+}z + 1. \tag{B.26}$$

Since this is a quadratic function with known roots that is only evaluated at integer inputs, if we want to bound the bias of $p_f$ it suffices to check the values of $\tilde{p}_f$ at the integers closest to the roots, namely

$\{\theta_1, \theta_1 + 1, \theta_2, \theta_2 + 1\}$. Plugging in these values gives

$$
\begin{aligned}
\tilde{p}_f(\theta_1) \quad &= 1 - \frac{\theta_2 - 1}{(1 + \frac{1}{2\theta_1})(\theta_2 + \frac{1}{2})} \\
&\geq 1 - \frac{1}{1 + \frac{1}{2\theta_1}} \\
&\geq \frac{1}{4\theta_1} \\
&\geq \frac{1}{4t},
\end{aligned}
\tag{B.27}
$$

where in the third line we used $\frac{1}{1+x} \leq 1 - x/2$ which holds for $0 \leq x \leq 1$. Using $\theta_2 \leq \theta_1 + 1$, we also have

$$
\begin{aligned}
\tilde{p}_f(\theta_1 + 1) \quad &= 1 - \frac{\theta_1 + 1}{\left(\theta_1 + \frac{1}{2}\right)\left(1 + \frac{1}{2\theta_2}\right)} \\
&\leq 1 - \frac{\theta_1 + 1}{\left(\theta_1 + \frac{1}{2}\right)\left(1 + \frac{1}{2\theta_1 + 2}\right)} \\
&= - \frac{1}{4\left(\theta_1 + \frac{1}{2}\right)\left(\theta_1 + \frac{3}{2}\right)} \\
&\leq - \frac{1}{4\left(t + \frac{1}{2}\right)\left(t + \frac{3}{2}\right)}.
\end{aligned}
\tag{B.28}
$$

$\tilde{p}_f(\theta_2)$ takes the same value as $\tilde{p}_f(\theta_1 + 1)$. Similarly,

$$
\begin{aligned}
\tilde{p}_f(\theta_2 + 1) \quad &= 1 - \frac{\theta_2 + 1}{(\theta_2 + \frac{1}{2})(1 + \frac{1}{2\theta_1})} \\
&\geq 1 - \frac{\theta_2 + 1}{(\theta_2 + \frac{1}{2})(1 + \frac{1}{2\theta_2 - 2})} \\
&= \frac{2\theta_2^2 - \frac{5}{4}}{\theta_2^2 - \frac{1}{4}}.
\end{aligned}
\tag{B.29}
$$

For $\theta_2 \leq 1$ this is a monotonically increasing function of $\theta_2$, and is thus lower bounded by picking $\theta_2 = 1$, giving $\tilde{p}_f(2) \geq 1$. It follows that for any choice of $t, \theta_1, \theta_2$, the bias is bounded from below by

$$
\beta' = \frac{1}{4\left(t + \frac{1}{2}\right)\left(t + \frac{3}{2}\right)}.
\tag{B.30}
$$

Note that our bound on the bias allows us to use the reduction from $f - \mathrm{BHP}_{N,t}$ to $\mathrm{QGNI}_{(t+1)N/t,t}$ for any valid choice of $f$ (satisfying eq. (B.24)).

$\square$

*Proof of Lemma 6.* Alice encodes her input in the quantum state

$$
|\tilde{X}\rangle_0 \equiv \frac{1}{\sqrt{2} \, \|X\|_F} |0\rangle \sum_{i=0}^{N-1} \sum_{j=0}^{D_0 - 1} X_{ij} |i, j\rangle + \frac{1}{\sqrt{2}} |1\rangle \left|0^{\otimes N}, 0^{\otimes D_0}\right\rangle
\tag{B.31}
$$

over $\log(ND_0) + 1$ qubits. She sends this state to Bob. Define $D = \max\{D_0, D_1\}$. Bob augments this state by attaching zero qubits and, reordering the first two qubits, obtains the state

$$
\begin{aligned}
|\tilde{X}\rangle \quad &\equiv \frac{1}{\sqrt{2} \, \|X\|_F} |0\rangle |0\rangle \sum_{i=0}^{N-1} \sum_{j=0}^{D_0 - 1} X_{ij} \left|i, j, 0^{\otimes(D - D_0)}\right\rangle + \frac{1}{\sqrt{2}} |1\rangle |0\rangle \left|0^{\otimes N}, 0^{\otimes D}\right\rangle \\
&\equiv \frac{1}{\sqrt{2}} |0\rangle |0\rangle |\overline{X}\rangle + \frac{1}{\sqrt{2}} |1\rangle |0\rangle \left|0^{\otimes N}, 0^{\otimes D}\right\rangle
\end{aligned}
\tag{B.32}
$$

over $\log(ND) + 2$ qubits.

Define by $\overline{W}_1$ the $D \times D$ matrix obtained by appending zero rows or columns to the rectangular matrix $W_1$ to obtain a square matrix, and denote $\alpha = \left\| A \otimes \overline{W}_1 \right\|$. Bob prepares an $(\alpha, 1, 0)$-block-encoding of $A \otimes \overline{W}_1$, denoted $U_{A \otimes \overline{W}_1}$, which acts on $\log(ND) + 1$ qubits. Bob then applies this unitary conditioned on the value of the first qubit, giving

$$
\begin{aligned}
\left( |0\rangle \langle 0| U_{A \otimes \overline{W}_1} + |1\rangle \langle 1| \right) |\tilde{X}\rangle \ &= \frac{1}{\sqrt{2}} |0\rangle U_{A \otimes \overline{W}_1} |0\rangle |\overline{X}\rangle + \frac{1}{\sqrt{2}} |1\rangle |0\rangle |0^{\otimes N}, 0^{\otimes D}\rangle \\
&= \frac{1}{\sqrt{2}} |0\rangle \left( \frac{1}{\alpha} |0\rangle A \otimes \overline{W}_1 |\overline{X}\rangle + |1\rangle |g\rangle \right) + \frac{1}{\sqrt{2}} |1\rangle |0\rangle |0^{\otimes N}, 0^{\otimes D}\rangle \\
&\equiv \frac{1}{\sqrt{2}} |0\rangle \left( \frac{1}{\alpha} |0\rangle |\overline{AXW_1}\rangle + |1\rangle |g\rangle \right) + \frac{1}{\sqrt{2}} |1\rangle |0\rangle |0^{\otimes N}, 0^{\otimes D}\rangle \\
&\equiv |\psi\rangle
\end{aligned}
$$

$$(B.33)$$

where $|g\rangle$ is an unnormalized garbage state. Above, $\overline{AXW_1}$ is an $N \times D$ matrix obtained by adding zero columns to $W_1$ as needed.

The sum pooling operator $\mathcal{P}$ can be implemented by multiplication by an $N/t \times N$ matrix which we denote by $\tilde{P}$. Define by $\overline{W}_2$ the $D \times N/t$ matrix obtained by appending zero rows to $W_2$ if needed. Given a matrix $M$ of size $N_1 \times N_2$, denote by $V[M]$ the vectorization of $M$. Bob then constructs the Hermitian matrix

$$
\mathcal{O} = \begin{pmatrix} 2a\alpha^2 |0\rangle \langle 0| \otimes \mathrm{diag}(V[\overline{W}_2 \tilde{P}]) & b\alpha |0\rangle \langle 0| \otimes V[\overline{W}_2 \tilde{P}] \langle 0^{\otimes N}, 0^{\otimes D}| \\ b\alpha |0\rangle \langle 0| \otimes |0^{\otimes N}, 0^{\otimes D}\rangle V[\overline{W}_2 \tilde{P}]^\dagger & 0 \end{pmatrix}. \quad (B.34)
$$

It follows that

$$
\begin{aligned}
\langle \psi | \mathcal{O} | \psi \rangle + c\,\mathrm{tr}\left[ W_2 P 1^{N \times D_1} \right] \ &= \frac{a}{\|X\|_F^2} \mathrm{tr}\left[ W_2 \tilde{P} \left( AXW_1 \right)^2 \right] + \frac{b}{\|X\|_F} \mathrm{tr}\left[ W_2 PAXW_1 \right] + c\,\mathrm{tr}\left[ W_2 P 1^{N \times D_1} \right] \\
&= \mathrm{tr}\left[ W_2 \tilde{P} \sigma(A \frac{X}{\|X\|_F} W_1) \right] \\
&= \varphi(X / \|X\|_F),
\end{aligned}
$$

$$(B.35)$$

where $1^{N \times D_1}$ is an all ones matrix. The last term on the RHS is independent of $X$ and can be computed by Bob without requiring Alice's message. Estimating $\langle \psi | \mathcal{O} | \psi \rangle$ to accuracy $\varepsilon$ requires $O(\|\mathcal{O}\| / \varepsilon)$ measurements. Since

$$
\begin{aligned}
\|\mathcal{O}\| \ &\leq \left\| 2a\alpha^2 |0\rangle \langle 0| \otimes \mathrm{diag}(V[\overline{W}_2 \tilde{P}]) \right\| + 2 \left\| b\alpha |0\rangle \langle 0| \otimes V[\overline{W}_2 \tilde{P}] \langle 0^{\otimes N}, 0^{\otimes D}| \right\| \\
&\leq 2(|a|\alpha^2 + |b|\alpha) \left\| W_2 \tilde{P} \right\|_\infty,
\end{aligned}
$$

$$(B.36)$$

Bob requires $O((|a|\alpha^2 + |b|\alpha) \left\| W_2 \tilde{P} \right\|_\infty / \varepsilon)$ copies of Alice's state in order to do this. $\qquad \square$

*Proof of lemma 7.* For the parameter choices used to obtain the classical lower bound (eq. (B.14) and eq. (B.16)), we have $\|W_1\| = 1, \|W_2 P\|_\infty \leq t$. Additionally, for the polynomials constructed in eq. (B.26), we have $|p_f(y)| \leq Ct^2$ from which it follows that the matrix $R$ used in the matrix representation of $p_f$ has constant operator norm $C$, and thus $\|A\| = \|S\| = \sqrt{C}$. We also have $a = 1, b = c = 0$ for the nonlinearity used (eq. (4.3)), and it thus follows from Lemma 6 that $Q_\varepsilon^\rightarrow(\mathrm{QGNI}_{N,t}) = O(t^3 \log(ND_0))$ for $\varepsilon \leq \frac{1}{4(t+\frac{1}{2})(t+\frac{3}{2})}$. With this choice of $\varepsilon$, the classical lower bound in Lemma 5 holds, and thus an exponential advantage in communication is obtained by using quantum communication. $\qquad \square$

*Proof of Lemma 11.* Consider first a single variable $z$, with data-dependent unitaries given by eq. (F.4a). If $\{\lambda_{\ell i}\}$ are chosen i.i.d. from a uniform distribution over say $[0, 1]$, then with probability 1 they are all unique and so are all sums of the form $\Lambda_{\overline{j}} = \sum_{\ell=1}^{L} \lambda_{\ell j_\ell}$ as well as differences $\Lambda_{\overline{j}} - \Lambda_{\overline{k}}$ for

$\overline{k} < \overline{j}$ where the inequality holds element-wise. Set $B_\ell$ to be the Hadamard transform over $\log N'$ qubits for all $\ell$, and pick the measurement operator $\mathcal{P}_0 = X_0$. We then have

$$
\begin{aligned}
\mathcal{L}_1 &= \langle\varphi|\,X_0\,|\varphi\rangle \\
&= \sum_{\overline{j},\overline{k}\in[N']^L} e^{2\pi i(\Lambda_{\overline{j}}-\Lambda_{\overline{k}})z}\left(B_1^\dagger\right)_{1j_1}\left(B_2^\dagger\right)_{j_1 j_2}\cdots\left(B_L^\dagger\right)_{j_{L-1}j_L}(X_0)_{j_L k_L}(B_L)_{k_L k_{L-1}}\cdots(B_1)_{k_1 1} \\
&= \sum_{\overline{j},\overline{k}\in[N']^L,\overline{j}\neq\overline{k}} e^{2\pi i(\Lambda_{\overline{j}}-\Lambda_{\overline{k}})z}\left(B_1^\dagger\right)_{1j_1}\left(B_2^\dagger\right)_{j_1 j_2}\cdots\left(B_L^\dagger\right)_{j_{L-1}j_L}(X_0)_{j_L k_L}(B_L)_{k_L k_{L-1}}\cdots(B_1)_{k_1 1} \\
&= \sum_{\overline{j}\in[N']^L}\sum_{\overline{k}<\overline{j}} 2\cos\left(2\pi(\Lambda_{\overline{j}}-\Lambda_{\overline{k}})z\right)\left(B_1^\dagger\right)_{1j_1}\cdots\left(B_L^\dagger\right)_{j_{L-1}j_L}(X_0)_{j_L k_L}\cdots(B_1)_{k_1 1} \\
&= \sum_{\overline{j}[:-1]\in[N']^{L-1}}\sum_{\overline{k}[:-1]<\overline{j}[:-1]}\sum_{j_L=1}^{N'} \frac{2\cos\left(2\pi(\Lambda_{\overline{j}[:-1]}-\Lambda_{\overline{k}[:-1]}+\lambda_{Lj_L}-\lambda_{L\tilde{j}_L})z\right)}{*\left(B_1^\dagger\right)_{1j_1}\cdots\left(B_L^\dagger\right)_{j_{L-1}j_L}(B_L)_{\tilde{j}_L,k_{L-1}}\cdots(B_1)_{k_1 1}}
\end{aligned}
$$
(B.37)

where $\tilde{j}_L = j_L + (-1)^{\lfloor j_L/(N'/2+1)\rfloor}N'/2$. In the third line, we dropped the diagonal terms in the double sum since they vanish due to the $X_0$ matrix having 0 on its diagonal. In the fourth line, we collected terms and used the symmetry of $\left(B_1^\dagger\right)_{1j_1}\cdots\left(B_L^\dagger\right)_{j_{L-1}j_L}(X_0)_{j_L k_L}\cdots(B_1)_{k_1 1}$ to the permutation of $\overline{j}$ and $\overline{k}$. In the last line we performed the sum over $k_L$ using the structure of $X_0$. By our assumption about the $\{\lambda_{\ell i}\}$, each term in the final sum has a unique frequency so no cancellations are possible. The coefficient of each cosine is nonzero (and is equal to $2N'^{-L}$ or $-2N'^{-L}$). There are a total of $\left(\frac{N'(N'-1)}{2}\right)^{L-1}N'$ such summands. This completes the first part of the proof for this choice of $\{B_\ell\}$.

Considering instead the case of two variables, with unitaries given by eq. (F.4b), an equivalent calculation gives

$$
\mathcal{L}_2 = \sum_{\overline{j}[:-1]\in[N']^{L-1}}\sum_{\overline{k}[:-1]<\overline{j}[:-1]}\sum_{j_L=1}^{N'} 2\cos\left(\omega_{\overline{j}\overline{k}}^1 y+\omega_{\overline{j}\overline{k}}^2 z\right)\left(B_1^\dagger\right)_{1j_1}\cdots\left(B_L^\dagger\right)_{j_{L-1}j_L}(B_L)_{\tilde{j}_L,k_{L-1}}\cdots(B_1)_{k_1 1},
$$
(B.38)

where

$$
\omega_{\overline{j}\overline{k}}^1 = 2\pi\left(\Lambda_{\overline{j}[:N'/2+1]}-\Lambda_{\overline{k}[:N'/2+1]}\right),\quad \omega_{\overline{j}\overline{k}}^2 = 2\pi\left(\Lambda_{\overline{j}[N'/2+1:-1]}-\Lambda_{\overline{k}[N'/2+1:-1]}+\lambda_{Lj_L}-\lambda_{L\tilde{j}_L}\right).
$$
(B.39)

As before, there are $\left(\frac{N'(N'-1)}{2}\right)^{L-1}N'$ summands in total. Since

$$
\cos\left(\omega_{\overline{j}\overline{k}}^1 y+\omega_{\overline{j}\overline{k}}^2 z\right) = \cos\left(\omega_{\overline{j}\overline{k}}^1 y\right)\cos\left(\omega_{\overline{j}\overline{k}}^2 z\right)-\sin\left(\omega_{\overline{j}\overline{k}}^1 y\right)\sin\left(\omega_{\overline{j}\overline{k}}^2 z\right),
$$
(B.40)

we can rewrite eq. (B.38) as a sum over $2\left(\frac{N'(N'-1)}{2}\right)^{L-1}N'$ terms that are pairwise orthogonal w.r.t. the $L^2$ inner product over $\mathbb{R}^2$. It follows from the definition of the separation rank that

$$
\mathrm{sep}\left(\mathcal{L}_2;y,z\right) = 2\left(\frac{N'(N'-1)}{2}\right)^{L-1}N'.
$$
(B.41)

We next use the assumption that the real and imaginary parts of each element of $B_\ell$ are real analytic function of parameters $\Theta$. This implies that the same property holds for product of entries of the form

$$
\left(B_1^\dagger\right)_{1j_1}\cdots\left(B_L^\dagger\right)_{j_{L-1}j_L}(B_L)_{\tilde{j}_L,k_{L-1}}\cdots(B_1)_{k_1 1}
$$
(B.42)

for any choice of $\overline{j},\overline{k}$. This coefficient is equal to 0 iff both the real and imaginary parts are equal to 0. Since the zero set of a real analytic function has measure 0 [81], the set of values of $\Theta$ for which any of the coefficients in eq. (B.38) vanishes also has measure 0, for all choices of $\overline{j},\overline{k}$. The result follows. $\qquad\square$

*Proof of Lemma 12.* Consider a periodic function $f$ with period 1. Denote by $S_M[f]$ the truncated Fourier series of $f$ written in terms of trigonometric functions:

$$
\begin{aligned}
S_M[f](y) &= \sum_{m=0}^{M-1} \int_{x=-1/2}^{1/2} f(x)\cos(2\pi mx)\,\mathrm{d}x \cos(2\pi my) + \sum_{m=0}^{M-1} \int_{x=-1/2}^{1/2} f(x)\sin(2\pi mx)\,\mathrm{d}x \sin(2\pi my) \\
&\equiv \sum_{m=0}^{M-1} \hat{f}_m^+ \cos(2\pi my) + \sum_{m=0}^{M-1} \hat{f}_m^- \sin(2\pi my).
\end{aligned}
\tag{B.43}
$$

If $f$ is $p$-times continuously differentiable, it is known that the Fourier series converges uniformly, with rate

$$
\|S_M[f] - f\|_\infty < \frac{C}{M^{p-1/2}}.
\tag{B.44}
$$

for some absolute constant $C$ [91]. For analytic functions the rate is exponential in $M$.

We now define the following circuit:

$$
A_1(x) = \mathrm{diag}((\underbrace{1,\ldots,1}_{N'/2}, \underbrace{1, e^{2\pi ix}, e^{2\pi i2x} \ldots, e^{2\pi i(N'/4-1)x}}_{N'/4}, \underbrace{1, e^{2\pi ix}, e^{2\pi i2x} \ldots, e^{2\pi i(N'/4-1)x}}_{N'/4})),
\tag{B.45}
$$

$$
B_1 = |\hat{f}\rangle\langle 0| + |0\rangle\langle\hat{f}|,
\tag{B.46}
$$

where

$$
|\hat{f}\rangle = \frac{1}{\sqrt{\sum_m |\hat{f}_m|}} \sum_{m=0}^{N'/4-1} \left( \sqrt{\hat{f}_m^+}\frac{|0\rangle + \mathrm{sign}(\hat{f}_m^+)|1\rangle}{\sqrt{2}}|0\rangle + \sqrt{\hat{f}_m^-}\frac{|0\rangle - i\mathrm{sign}(\hat{f}_m^-)|1\rangle}{\sqrt{2}}|1\rangle \right)|m\rangle.
\tag{B.47}
$$

Choosing $|\psi(x)\rangle = |0\rangle$ as the initial state, this gives

$$
\begin{aligned}
|\varphi\rangle &= A_1 B_1 |0\rangle \\
&= A_1 |\hat{f}\rangle \\
&= \frac{1}{\sqrt{\sum_m |\hat{f}_m|}} \sum_{m=0}^{N'/4-1} \left( \sqrt{\hat{f}_m^+}\frac{|0\rangle + \mathrm{sign}(\hat{f}_m^+)e^{2\pi imx}|1\rangle}{\sqrt{2}}|0\rangle + \sqrt{\hat{f}_m^-}\frac{|0\rangle - i\mathrm{sign}(\hat{f}_m^-)e^{2\pi imx}|1\rangle}{\sqrt{2}}|1\rangle \right)|m\rangle
\end{aligned}
\tag{B.48}
$$

It follows that

$$
\begin{aligned}
\langle\varphi|X_0|\varphi\rangle &= \frac{1}{\sum_m |\hat{f}_m|} \sum_{m=0}^{N'/4-1} 
\begin{array}{l}
\left|\hat{f}_m^+\right| \frac{\langle 0| + \mathrm{sign}(\hat{f}_m^+)e^{-2\pi imx}\langle 1|}{\sqrt{2}} X_0 \frac{|0\rangle + \mathrm{sign}(\hat{f}_m^+)e^{2\pi imx}|1\rangle}{\sqrt{2}} \\
+ \left|\hat{f}_m^-\right| \frac{\langle 0| + i\mathrm{sign}(\hat{f}_m^-)e^{-2\pi imx}\langle 1|}{\sqrt{2}} X_0 \frac{|0\rangle - i\mathrm{sign}(\hat{f}_m^-)e^{2\pi imx}|1\rangle}{\sqrt{2}}
\end{array} \\
&= \frac{1}{\sum_m |\hat{f}_m|} \sum_{m=0}^{N'/4-1} \hat{f}_m^+ \cos(2\pi mx) + \hat{f}_m^- \sin(2\pi mx) \\
&= \frac{1}{\sum_m |\hat{f}_m|} S_{N'/4}[f](x)
\end{aligned}
\tag{B.49}
$$

This approximation thus converges uniformly according to eq. (B.44), with error decaying exponentially with number of qubits $\log N'$ as long as $f$ is continuously differentiable at least once. $\square$

*Proof of Lemma 10.* The algorithm in Theorem 5 of [100] takes as input a state-preparation unitary $U$ acting on $n = \log N$ qubits such that $U|0\rangle^{\otimes n} = |z\rangle$. Using $O(\log 1/\varepsilon)$ queries to $U$ and $U^\dagger$ and

$n + 4$ ancillas, it creates a state $|\varphi\rangle$ such that measuring 0 on the first $n + 4$ qubits of $|\varphi\rangle$ results in a state $|\hat{\varphi}\rangle$ that obeys

$$\left\| |\hat{\varphi}\rangle - \frac{1}{\|\sigma(z)\|_2} |\sigma(z)\rangle \right\|_2 < \varepsilon. \tag{B.50}$$

Additionally, the probability of measuring 0 on the first $n + 4$ qubits is $O(1)$.

We will be interested in applying this algorithm to the state $|U_1 x\rangle$. The state preparation unitary can be instantiated with a single round of communication by Alice starting with the state $|0\rangle^{\otimes 2n+4}$, applying a unitary that encodes $x$ in the last $n$ qubits of this state, and then sending it to Bob who applies $U_1$ to the same $n$ qubits. The conjugate of the state-preparation unitary can be applied in a similar fashion by reversing this procedure. This can include any conditioning required on the values of the other qubits.

Based on the query complexity of the algorithm in [100] to the state preparation unitary, $O(\log(1/\varepsilon))$ rounds will suffice to obtain a state

$$|\tilde{\varphi}_\sigma\rangle = \alpha |0\rangle^{\otimes n+4} |\tilde{y}\rangle + |\phi\rangle, \tag{B.51}$$

such that

$$\left\| |\tilde{y}\rangle - \left| \frac{1}{\|\sigma(U_1 x)\|_2} \sigma(U_1 x) \right\rangle \right\|_2 < \varepsilon. \tag{B.52}$$

Bob then applies $U_2$ to the state $|\tilde{\varphi}_\sigma\rangle$ conditioned on the first $n + 4$ qubits being in the state $|0\rangle^{\otimes n+4}$. The state $|\phi\rangle$ is unaffected. Unitary of $U_2$ combined with the above bound guarantees

$$\left\| |\hat{y}\rangle - \left| U_2 \frac{1}{\|\sigma(U_1 x)\|_2} \sigma(U_1 x) \right\rangle \right\|_2 < \varepsilon. \tag{B.53}$$

Additionally, from Theorem 3 of [100] we are guaranteed that $\alpha = O(1)$. $\qquad\square$

## C   Data parallelism

Data parallelism involves storing multiple copies of a model on different devices and training each copy on a subset of the full data. We consider a model of the form

$$|\varphi(\Theta, x)\rangle \equiv \left( \prod_{\ell=L}^{1} U_\ell(\theta_\ell, x) \right) |x\rangle, \tag{C.1}$$

where $x$ is an $N_1 \times N_2$ matrix which we write as $x = [x_A, x_B]$ for two $N_1/2 \times N_2$ matrices $x_A, x_B$. Assume also that $\|x\|_F = 1$. This model can be used to define a distributed problem with dara parallelism by considering the following inputs to both players:

$$\begin{aligned} \text{Alice}: & \quad x_A, \{U_\ell\}, \\ \text{Bob}: & \quad x_B, \{U_\ell\}. \end{aligned} \tag{C.2}$$

The state $|x\rangle$ can be prepared in a single round of communication involving $\log(N_1 N_2)$ qubits. Alice simply prepares the state

$$\begin{aligned} |x_A\rangle + \sqrt{1 - \|x_A\|_F^2} \, |N_1/2, 0\rangle \quad &= (x_A)_{ij} \sum_{i=0}^{N_1/2-1} \sum_{j=0}^{N_2-1} |i, j\rangle + \sqrt{1 - \|x_A\|_F^2} \, |N_1/2, 0\rangle \\ &= (x_A)_{ij} \sum_{i=0}^{N_1/2-1} \sum_{j=0}^{N_2-1} |i, j\rangle + \|x_B\|_F \, |N_1/2, 0\rangle, \end{aligned} \tag{C.3}$$

using zero-based indexing of the elements of $x_A$. After sending this to Bob, he applies the unitary

$$\frac{1}{\|x_B\|_F} (x_B)_{i,j} \sum_{i=0}^{N_1/2-1} \sum_{j=0}^{N_2-1} |ij\rangle \langle N_1/2, 0| + h.c.. \tag{C.4}$$

The resulting state is $|x\rangle$. As before, the gradients with respect to the parameters of the unitaries $\{U_\ell\}$ can be estimated by preparing copies of this state and using shadow tomography. The number of copies will again be logarithmic in $N_1, N_2$ and the number of trainable parameters.

# D    Exponential advantages in end-to-end training

So far we have discussed the problems of inference and estimating a single gradient vector. It is natural to also consider when these or other gradient estimators can be used to efficiently solve an optimization problem (i.e. when the entire training processes is considered rather than a single iteration). Applying the gradient estimation algorithm detailed in Lemma 2 iteratively gives a distributed stochastic gradient descent algorithm which we detail in Algorithm 2, yet one may be concerned that a choice of $\varepsilon = O(\log N)$ which is needed to obtain an advantage in communication complexity will preclude efficient convergence. Here we present a simpler algorithm that requires a single quantum measurement per iteration, and can provably solve certain convex problems efficiently, as well as an application of shadow tomography to fine-tuning where convergence can be guaranteed, again with only logarithmic communication cost. In both cases, there is an exponential advantage in communication even when considering the entire training process.

## D.1    "Smooth" circuits

Consider the case where $A_\ell$ are product of rotations for all $\ell$, namely

$$A_\ell = \prod_{j=1}^{P} e^{-\frac{1}{2}i\beta_{\ell j}^A \theta_{\ell j}^A \mathcal{P}_{\ell j}^A}, \tag{D.1}$$

where $\mathcal{P}_{\ell j}^A$ are Pauli matrices acting on all qubits, and similarly for $B_\ell$. These can also be interspersed with other non-trainable unitaries. This constitutes a slight generalization of the setting considered in [49], and the algorithm we present is essentially a distributed distributed version of theirs. Denote by $\beta$ an $2PL$-dimensional vector with elements $\beta_{\ell j}^Q$ where $Q \in \{A, B\}$ [9]. The quantity $\|\beta\|_1$ is the total evolution time if we interpret the state $|\varphi\rangle$ as a sequence of Hamiltonians applied to the initial state $|x\rangle$.

In Appendix D.3 we describe an algorithm that converges to the neighborhood of a minimum, or achieves $\mathbb{E}\mathcal{L}(\Theta) - \mathcal{L}(\Theta^\star) \leq \varepsilon_0$, for a convex $\mathcal{L}$ after

$$2 \left\|\Theta^{(0)} - \Theta^\star\right\|_2^2 \|\beta\|_1^2 / \varepsilon_0^2 \tag{D.2}$$

iterations, where $\Theta^\star$ are the parameter values at the minimum of $\mathcal{L}$. The expectation is with respect to the randomness of quantum measurement and additional internal randomness of the algorithm. The algorithm is based on classically sampling a single coordinate to update at every iteration, and computing an unbiased estimator of the gradient with a single measurement. It can thus be seen as a form of probabilistic coordinate descent.

This implies an exponential advantage in communication for the entire training process as long as $\left\|\Theta^{(0)} - \Theta^\star\right\|_2^2 \|\beta\|_1^2 = \mathrm{polylog}(N)$. Such circuits either have a small number of trainable parameters ($P = O(\mathrm{polylog}(N))$), depend weakly on each parameter (e.g. $\beta_{\ell j}^Q = O(1/P)$ for arbitrary $P$), or have structure that allows initial parameter guesses whose quality diminishes quite slowly with system size. Nevertheless, over a convex region the loss can rapidly change by an $O(1)$ amount. One may also be concerned that in the setting $\left\|\Theta^{(0)} - \Theta^\star\right\|_2^2 \|\beta\|_1^2 = \mathrm{polylog}(N)$ only a logarithmic number of parameters is updated during the entire training process and so the total effect of the training process may be negligible. It is important to note however that each such sparse update depends on the structure of the entire gradient vector as seen in the sampling step. In this sense the algorithm is a form of probabilistic coordinate descent, since the probability of updating a coordinate $|\beta_{\ell j}^Q|/\|\beta\|_1$ is proportional to the the magnitude of the corresponding element in the gradient (actually serving as an upper bound for it).

Remarkably, the time complexity of a single iteration of this algorithm is proportional to a forward pass, and so matches the scaling of classical backpropagation. This is in contrast to the polynomial overhead of shadow tomography (Theorem 1). Additionally, it requires a single measurement per iteration, without any of the additional factors in the sample complexity of shadow tomography.

---

[9][49] actually consider a related quantity for which has smaller norm in cases where multiple gradient measurements commute, leading to even better rates.

## D.2 Fine-tuning the last layer of a model

Consider a model given by eq. (3.1) where only the parameters of $A_L$ are trained, and the rest are frozen, and denote this model by $|\varphi_f\rangle$. The circuit up to that unitary could include multiple data-dependent unitaries that represent complex features in the data. Training only the final layer in this manner is a common method of fine-tuning a pre-trained model [53]. If we now define

$$\tilde{E}^A_{Li} = |1\rangle\langle 0| \otimes A_L^\dagger \mathcal{P}_0 \frac{\partial A_L}{\partial \theta^A_{Li}} + |0\rangle\langle 1| \otimes \left(\frac{\partial A_L}{\partial \theta^A_{Li}}\right)^\dagger \mathcal{P}_0 A_L, \tag{D.3}$$

the expectation value of $\tilde{E}^A_{Li}$ using the state $|+\rangle\,|\mu^A_L\rangle$ gives $\frac{\partial \mathcal{L}}{\partial \theta^A_{\ell i}}$. Here $|\mu^A_L\rangle = B_L(x)\prod_{k=L-1}^{1} A_k(x)B_k(x)\,|\psi(x)\rangle$ is the forward feature computed by Alice at layer $L$ with the parameters of all the other unitaries frozen (hence the dependence on them is dropped). Since the observables in the shadow tomography problem can be chosen in an online fashion [5, 6, 13], and adaptively based on previous measurements, we can simply define a stream of measurement operators by measuring $P$ observables to estimate the gradients w.r.t. an initial set of parameters, updating these parameters using gradient descent with step size $\eta$, and defining a new set of observables using the updated parameters. Repeating this for $T$ iterations gives a total of $PT$ observables (a complete description of the algorithm is given in Algorithm 3).

By the scaling in Lemma 2, the total communication needed is $\tilde{O}(\log N(\log TP)^2 \log(1/\delta)/\varepsilon^4)$ over $O(L)$ rounds (since only $O(L)$ rounds are needed to create copies of $|\mu^{A_L}\rangle$). This implies an exponential advantage in communication for the entire training process (under the reasonable assumption $T = O(\mathrm{poly}(N, P))$), despite the additional stochasticity introduced by the need to perform quantum measurements. For example, assume one has a bound $\|\nabla\mathcal{L}\|_2^2 \leq K$. If the circuit is comprised of unitaries with Hermitian derivatives, this holds with $K = PL$. In that case, denoting by $g$ the gradient estimator obtained by shadow tomography, we have

$$\|g\|_2^2 \leq \|\nabla\mathcal{L}\|_2^2 + \|\nabla\mathcal{L} - g\|_2^2 \leq K + \varepsilon^2 PL. \tag{D.4}$$

It then follows directly from Lemma 8 that for an appropriately chosen step size, if $\mathcal{L}$ is convex one can find parameter values $\overline{\Theta}$ such that $\mathcal{L}(\overline{\Theta}) - \mathcal{L}(\Theta^\star) \leq \varepsilon_0$ using

$$T = 2\left\|\Theta^{(0)} - \Theta^\star\right\|_2^2 (K + \varepsilon^2 PL)^2/\varepsilon_0^2 \tag{D.5}$$

iterations of gradient descent. Similarly if $\mathcal{L}$ is $\lambda$-strongly convex then $T = 2(K + \varepsilon^2 PL)^2/\lambda\varepsilon_0 + 1$ iterations are sufficient. In both cases therefore an exponential advantage is achieved for the optimization process as a whole, since in both cases one can implement the circuit that is used to obtain the lower bounds in Lemma 3.

In the following, we make use of well-known convergence rates for stochastic gradient descent:

**Lemma 8** ([26]). *Given an objective function $\mathcal{L}(\Theta)$ with a minimum at $\Theta^\star$ and a stochastic gradient oracle that returns a noisy estimate of the gradient $g(\Theta)$ such that $\mathbb{E}g(\Theta) = \nabla\mathcal{L}(\Theta), \mathbb{E}\|g\|_2^2 \leq G^2$, and denoting by $\Theta^{(0)}$ a point in parameter space and $R = \left\|\Theta^{(0)} - \Theta^\star\right\|_2$, we have:*

*i) If $\mathcal{L}$ is convex in a Euclidean ball of radius $R$ around $\Theta^\star$, then gradient descent with step size $\eta = \frac{R}{G}\sqrt{\frac{2}{T}}$ achieves*

$$\mathbb{E}\mathcal{L}(\frac{1}{T}\sum_{t=1}^{T}\Theta^{(t)}) - \mathcal{L}(\Theta^\star) \leq RG\sqrt{\frac{2}{T}}. \tag{D.6}$$

*ii) If $\mathcal{L}$ is $\lambda$-strongly convex in a Euclidean ball of radius $R$ around $\Theta^\star$, then gradient descent with step size $\eta_t = \frac{2}{\lambda(t+1)}$ achieves*

$$\mathbb{E}\mathcal{L}(\frac{1}{T(T+1)}\sum_{t=1}^{T}2t\Theta^{(t)}) - \mathcal{L}(\Theta^\star) \leq \frac{2G^2}{\lambda(T+1)}. \tag{D.7}$$

---

**Algorithm 1** Distributed Probabilistic Coordinate Descent

---

**Input:** Alice: $x, \{A_\ell\}, \Theta_A, \{\eta_t\}, T$. Bob: $\{B_\ell\}, \Theta_B, \{\eta_t\}, T$.

**Output:** Alice: Updated parameters $\Theta_A^{(T)}$. Bob: Updated parameters $\Theta_B^{(T)}$.

  1: Alice and Bob each pre-process their coefficient vectors $\beta_A, \beta_B$ to enable efficient sampling.
  2: Alice sends $\|\beta_A\|_1$ to Bob. {$O(\log P)$ bits of classical communication.}
  3: **for** $t \in \{1, \ldots, T\}$ **do**
  4:     Bob samples $b \sim \text{Bernoulli}(\|\beta_A\|_1 / \|\beta\|_1)$ and sends $b$ to Alice {1 bit of classical communi-
       cation.}
  5:     **if** $b == 0$ **then**
  6:         Bob samples $(\ell, i)$ from the discrete distribution defined by $\text{abs}(\beta_B)$
  7:         Bob create the state $\left|\psi_{\ell 0}^B\right\rangle$ {$O(L)$ rounds of quantum communication}
  8:         Bob measures $\hat{E}_{\ell i}^B$, as defined in eq. (D.8), obtaining a result $m \in \{-1, 1\}$
  9:         Bob sets $\theta_{\ell i}^B \leftarrow \theta_{\ell i}^B - \eta_t \text{sign}(\beta_{\ell i}^B)\|\beta\|_1 m$
10:     **else**
11:         Alice runs steps 6-9, (replacing $B$ with $A$)
12:     **end if**
13: **end for**

---

## D.3   Distributed Probabilistic Coordinate Descent

Given distributed states of the form eq. (D.1), optimization over $\Theta$ can be performed using Algorithm 1. We verify the correctness of this algorithm and provide convergence rates following [49]. Define the Hermitian measurement operator

$$\hat{E}_{\ell i}^Q = \left(|0\rangle \langle 0| \otimes I - i |1\rangle \langle 1| \otimes \mathcal{P}_{\ell i}^Q\right)^\dagger X_a \left(|0\rangle \langle 0| \otimes I - i |1\rangle \langle 1| \otimes \mathcal{P}_{\ell i}^Q\right), \qquad \text{(D.8)}$$

with eigenvalues in $\{-1, 1\}$. Note that $\beta_{\ell i}^Q \left\langle \psi_{\ell 0}^Q\right| \hat{E}_{\ell i}^Q \left|\psi_{\ell 0}^Q\right\rangle = \frac{\partial \mathcal{L}}{\partial \theta_{\ell i}^Q}$, and this is essentially a compact way of representing a Hadamard test for the relevant expectation value. Now consider a gradient estimator that first samples $(Q, \ell, i)$ with probability $|\beta_{\ell i}^Q|/\|\beta\|_1$, then returns a one-sparse vector with $g_{\ell i}^Q = \text{sign}(\beta_{\ell i}^Q) \|\beta\|_1 m$, where $m$ is the result of a single measurement of $\hat{E}_{\ell i}^Q$ using the state $\left|\psi_{\ell 0}^Q\right\rangle$. For this estimator we have

$$\mathbb{E}g_{\ell i}^Q = \text{sign}(\beta_{\ell i}^Q) \|\beta\|_1 \frac{\left|\beta_{\ell i}^Q\right|}{\|\beta\|_1} \left\langle \psi_{\ell 0}^A\right| \hat{E}_{\ell i}^Q \left|\psi_{\ell 0}^A\right\rangle = \frac{\partial \mathcal{L}}{\partial \theta_{\ell i}^Q}, \qquad \text{(D.9)}$$

where the expectation is taken over both the index sampling process and the quantum measurement. The procedure generates a valid gradient estimator.

In order to show convergence, one simply notes that by construction, $\|g\|_2 = \|\beta\|_1$. It then follows immediately from Lemma 8 that, with an appropriately chosen step size, Algorithm 1 achieves $\mathbb{E}\mathcal{L}(\Theta) - \mathcal{L}(\Theta^\star) \leq \varepsilon_0$ for a convex $\mathcal{L}$ using

$$\frac{2 \left\|\Theta^{(0)} - \Theta^\star\right\|_2^2 \|\beta\|_1^2}{\varepsilon_0^2} \qquad \text{(D.10)}$$

queries. For a $\lambda$-strongly convex $\mathcal{L}$, only

$$\frac{2 \|\beta\|_1^2}{\lambda \varepsilon_0} + 1 \qquad \text{(D.11)}$$

queries are required. The pre-processing in step 1 of Algorithm 1 requires time $O(P \log P)$ and subsequently enables sampling in time $O(1)$ using e.g. [111] [10].

---

**Algorithm 2** Shadow Tomographic Distributed Gradient Descent

---

**Input:** Alice: $x, \{A_\ell\}, \Theta_A^{(1)}, \eta, T$. Bob: $\{B_\ell\}, \Theta_B^{(1)}, \eta, T$.

**Output:** Alice: Updated parameters $\Theta_A^{(T)}$. Bob: Updated parameters $\Theta_B^{(T)}$.

1: **for** $t \in \{1, \ldots, T\}$ **do**
2:     **for** $\ell \in \{1, \ldots, L\}$ **do**
3:         Alice prepares $\tilde{O}(\log^2 P \log N' \log(L/\delta)/\varepsilon^4)$ copies of $\left|\psi_{\ell 0}^A(\Theta^{(t)})\right\rangle$ $\{O(L)$ rounds of communication$\}$
4:         Alice runs Shadow Tomography to estimate $\{\mathbb{E}E_{\ell i}^A(\Theta^{(t)})\}_{i=1}^P$ up to error $\varepsilon$, denoting these $\{g_{\ell i}^A(\Theta^{(t)})\}_{i=1}^P$.
5:         Bob prepares $\tilde{O}(\log^2 P \log N' \log(L/\delta)/\varepsilon^4)$ copies of $\left|\psi_{\ell 0}^B(\Theta^{(t)})\right\rangle$ $\{O(L)$ rounds of communication$\}$
6:         Bob runs Shadow Tomography to estimate $\{\mathbb{E}E_{\ell i}^B(\Theta^{(t)})\}_{i=1}^P$ up to error $\varepsilon$, denoting these $\{g_{\ell i}^B(\Theta^{(t)})\}_{i=1}^P$.
7:         Alice sets $\theta_\ell^{A(t+1)} \leftarrow \theta_\ell^{A(t)} - \eta g_\ell^A(\Theta^{(t)})$.
8:         Bob sets $\theta_\ell^{B(t+1)} \leftarrow \theta_\ell^{B(t)} - \eta g_\ell^B(\Theta^{(t)})$.
9:     **end for**
10: **end for**

---

---

**Algorithm 3** Shadow Tomographic Distributed Fine-Tuning

---

**Input:** Alice: $x, \{A_\ell\}, \theta_L^{A(1)}, \eta, T$. Bob: $\{B_\ell\}$

**Output:** Alice: Updated parameters $\Theta_A^{(T)}$.

1: Alice prepares $\tilde{O}(\log^2(PT) \log N' \log(1/\delta)/\varepsilon^4)$ copies of $\left|\mu_L^A\right\rangle$ $\{O(L)$ rounds of communication$\}$
2: **for** $t \in \{1, \ldots, T\}$ **do**
3:     Alice runs online Shadow Tomography to estimate $\{\mathbb{E}\tilde{E}_{Li}^A(\theta_L^{A(t)})\}$ up to error $\varepsilon$, denoting these $\{g_{Li}^A(\theta_L^{A(t)})\}$.
4:     Alice sets $\theta_L^{A(t+1)} \leftarrow \theta_L^{A(t)} - \eta g_L^A(\theta_L^{A(t)})$.
5: **end for**

---

### D.4 Algorithms based on Shadow Tomography

## E Communication Complexity of Linear Classification

While the separation in communication complexity for expressive networks can be quite large, interestingly we will show that for some of the simplest models this advantage can vanish due to the presence of structure. In particular, when a linear classifier is well-suited to a task such that the margin is large, the communication advantage will start to wane, while a lack of structure in linear classification will make the problem difficult for quantum algorithms as well. More specifically, we consider the following classification problem:

**Problem 7** (Distributed Linear Classification). *Alice and Bob are given $x, y \in S^N$, with the promise that $|x \cdot y| \geq \gamma$ for some $0 \leq \gamma \leq 1$. Their goal is to determine the sign of $x \cdot y$.*

This is one of the simplest distributed inference problem in high dimensions that one can formulate. $x$ can be thought of as the input to the model, while $y$ defines a separating hyperplane with some margin. Since with finite margin we are only required to resolve the inner product between the vectors to some finite precision, it might seem that an exponential quantum advantage should be possible for this problem by encoding the inputs in the amplitudes of a quantum state. However, we show that classical algorithms can leverage this structure as well, and consequently that the quantum advantage in communication that can be achieved for this problem is at most polynomial in $N$. We

---

[10]An even simpler algorithm that sorts the lists as a pre-processing step and uses inverse CDF sampling will enable sampling with cost $O(\log P)$

prove this with respect to the the randomized classical communication model, in which Alice and Bob are allowed to share random bits that are independent of their inputs [11].

**Lemma 9.** *The quantum communication complexity of Problem 7 is $\Omega\left(\sqrt{N/\max(1, \lceil \gamma N \rceil)}\right)$. The randomized classical communication complexity of Problem 7 is $O(\min(N, 1/\gamma^2))$.*

*Proof.* We first describe a protocol that allows Alice and Bob to solve the linear classification problem with margin $\gamma$ using $O(1/\gamma^2)$ bits of classical communication and shared randomness, assuming $\gamma > 0$. Note that this bound accords with the notion that the margin rather than the ambient dimension sets the complexity of these types of problems, which is also manifest in the sample complexity of learning with linearly separable data.

Alice and Bob share $kN$ bits sampled i.i.d. from a uniform distribution over $\{0, 1\}$, and that these bits are arranged in a $k \times N$ matrix $R$. Alice and Bob then receive $x$ and $y$ respectively, which are valid inputs to the linear classification problem with margin $\gamma$. For any $N$-dimensional vector $z$, define the random projection

$$f : \mathbb{R}^n \to \mathbb{R}^k, \quad f(z) = \frac{1}{k}(2R - 1)z, \tag{E.1}$$

where addition is element-wise. Applying the Johnson-Lindenstrauss lemma for projections with binary variables [8], we obtain that if $k = C/\varepsilon^2$, for some absolute constant $C$, then with probability larger than $2/3$ we have for any $z, z' \in \{x, y, 0\}$ (all of these being vectors in $\mathbb{R}^N$), $f$ is an approximate isometry in the sense

$$(1 - \varepsilon) \|z - z'\|_2^2 \le \|f(z) - f(z')\|_2^2 \le (1 + \varepsilon) \|z - z'\|_2^2. \tag{E.2}$$

The key feature of this result is that $k$ is completely independent of $N$. Applying it repeatedly gives

$$\begin{aligned} \|f(x) - f(y)\|_2^2 - \|f(x)\|_2^2 - \|f(y)\|_2^2 &\le (1 + \varepsilon) \|x - y\|_2^2 - 2(1 - \varepsilon) \\ f(x) \cdot f(y) &\ge (1 + \varepsilon)x \cdot y - 2\varepsilon. \end{aligned} \tag{E.3}$$

Obtaining an upper bound in a similar fashion using the converse inequalities, we have

$$(1 + \varepsilon)x \cdot y - 2\varepsilon \le f(x) \cdot f(y) \le (1 - \varepsilon)x \cdot y + 2\varepsilon. \tag{E.4}$$

Assume now that $x, y$ are valid inputs to the linear classification problem with margin $\gamma$, and specifically that $x \cdot y \ge \gamma$. The lower bound above gives

$$(1 + \varepsilon)\gamma - 2\varepsilon \le f(x) \cdot f(y), \tag{E.5}$$

and if we choose $\varepsilon = \gamma/8$ we obtain

$$\frac{\gamma}{2} \le (1 + \frac{\gamma}{8})\gamma - \frac{\gamma}{4} \le f(x) \cdot f(y), \tag{E.6}$$

where we used $\gamma \le 1$. Similarly, if instead $x \cdot y \le -\gamma$ we obtain

$$f(x) \cdot f(y) \le -(1 - \frac{\gamma}{8})\gamma + \frac{\gamma}{4} \le -\frac{\gamma}{2}. \tag{E.7}$$

It follows that if Alice computes $f(x)$ and sends the resulting $O(k) = O(1/\gamma^2)$ bits that describe this vector to Bob (assuming some finite precision that is large enough so as not to affect the margin, which will contribute ), Bob can simply compute $f(x) \cdot f(y)$ which will reveal the result of the classification problem, which he can then communicate to Alice using a single bit.

If $\gamma = 0$ there is a trivial $O(N)$ classical algorithm where Alice sends Bob $x$.

We next describe the quantum lower bound for Problem 7. Denote by $d_H$ the Hamming distance between binary vectors. We will use lower bounds for the following problem:

**Problem 8** (Gap Hamming with general gap). *Alice and Bob are given $\hat{x}, \hat{y} \in \{0, 1\}^N$ respectively. Given a promise that either $d_H(\hat{x}, \hat{y}) \ge N/2 + g/2$ or $d_H(\hat{x}, \hat{y}) \le N/2 - g/2$, Alice and Bob must determine which one is the case.*

---

[11]This resource can have a dramatic effect on the communication complexity of a problem. The canonical example is equality of $N$ bit strings, which can be solved with constant success probability using 1 bit of communication and shared randomness, while requiring $N$ bits of communication otherwise.

There is a simple reduction from Problem 8 to Problem 7 for certain values of $\gamma$, which we will then use to obtain a result for all $\gamma$. Assuming Alice is given $\hat{x}$ and Bob is given $\hat{y}$, they construct unit norm real vectors by $x = (2\hat{x} - 1)/\sqrt{N}, y = (2\hat{y} - 1)/\sqrt{N}$ with addition performed element-wise. If $d_H(x, y) \geq N/2 + g/2$ then

$$
\begin{aligned}
x \cdot y &= \sum_{i, x_i = y_i} \frac{1}{N} + \sum_{i, x_i \neq y_i} (-\frac{1}{N}) \\
&\geq \frac{N+g}{2} \frac{1}{N} + \frac{N-g}{2} (-\frac{1}{N}) \\
&= \frac{g}{N}.
\end{aligned}
\tag{E.8}
$$

Similarly, $d_H(\hat{x}, \hat{y}) \leq N/2 - g/2 \Rightarrow x \cdot y \leq -g/N$. It follows that $x, y$ are valid inputs for a linear classification problem over the unit sphere with margin $2g/N$. From the results of [88], any quantum algorithm for the Gap Hamming problem with gap $g \in \{1, \ldots, N\}$ requires $\Omega(\sqrt{N/g})$ qubits of communication. It follows that the linear classification problem requires $\Omega(\sqrt{1/\gamma})$ qubits of communication. This bound holds for integer $\gamma N$. To get a result for general $0 < \gamma \leq 1$ we simply note that the communication complexity must be a non-decreasing function of $1/\gamma$, since any inputs which constitute a valid instance with some $\gamma$ are also a valid instance for any $\gamma' < \gamma$. Given some real $\gamma$, the resulting communication problem is at least as hard as the one with margin $\lceil \gamma N \rceil / N \geq \gamma$. It follows that a $\Omega(\sqrt{N/\lceil \gamma N \rceil})$ bound holds for all $0 < \gamma \leq 1$.

If $\gamma = 0$, by a similar argument we can apply the lower bound for $\gamma = 1/N$, implying that $\Omega(\sqrt{N})$ qubits of communication are necessary. Once again there is only a polynomial advantage at best. $\square$

## F  Expressivity of quantum circuits

### F.1  Expressivity of compositional models

It is natural to ask how expressive models of the form of eq. (3.1) can be, given the unitarity constraint of quantum mechanics on the matrices $\{A_\ell, B_\ell\}$. This is a nuanced question that can depend on the encoding of the data that is chosen and the method of readout. On the one hand, if we pick $|\psi(x)\rangle$ as in eq. (3.4) and use $\{A_\ell, B_\ell\}$ that are independent of $x$, the resulting state $|\varphi\rangle$ will be a linear function of $x$ and the observables measured will be at most quadratic functions of those entries. On the other hand, one could map bits to qubits 1-to-1 and encode any reversible classical function of data within the unitary matrices $\{A_\ell(x)\}$ with the use of extra space qubits. However, this negates the possibility of any space or communication advantages (and does not provide any real computational advantage without additional processing). As above, one prefers to work on more generic functions in the amplitude and phase space, allowing for an exponential compression of the data into a quantum state, but one that must be carefully worked with.

We investigate the consequences of picking $\{A_\ell(x)\}$ that are *nonlinear* functions of $x$, and $\{B_\ell\}$ that are data-independent. This is inspired by a common use case in which Alice holds some data or features of the data, while Bob holds a model that can process these features. Given a scalar variable $x$, define $A_\ell(x) = \text{diag}((e^{-2\pi i \lambda_{\ell 1} x}, \ldots, e^{-2\pi i \lambda_{\ell N'} x}))$ for $\ell \in \{1, \ldots, L\}$. We also consider parameterized unitaries $\{B_\ell\}$ that are independent of the $\{\lambda_{\ell i}\}$ and inputs $x, y$, and the state obtained by interleaving the two in the manner of eq. (3.1) by $|\varphi(x)\rangle$.

We next set $\lambda_{\ell 1} = 0$ for all $\ell \in \{1, \ldots, L\}$ and $\lambda_{L2} = 0$. If we are interested in expressing the frequency

$$
\Lambda_{\bar{j}} = \sum_{\ell=1}^{L-1} \lambda_{\ell j_\ell},
\tag{F.1}
$$

where $j_\ell \in \{2, \ldots, N'\}$, we simply initialize with $|\psi(x)\rangle = |+\rangle_0 |0\rangle$ and use

$$
B_\ell = |j_\ell - 1\rangle \langle j_{\ell-1} - 1| + |j_{\ell-1} - 1\rangle \langle j_\ell - 1|,
\tag{F.2}
$$

with $j_1 = j_L = 2$. It is easy to check that the resulting state is $|\varphi(x)\rangle = (|0\rangle + e^{-2\pi i \Lambda_{\bar{j}} x} |1\rangle)/\sqrt{2}$. Since the basis state $|0\rangle$ does not accumulate any phase, while the $B_\ell$s swap the $|1\rangle$ state with

the appropriate basis state at every layer in order to accumulate a phase corresponding to a single summand in eq. (F.1). Choosing to measure the operator $\mathcal{P}_0 = X_0$, it follows that $\langle \varphi(x) | \, X_0 \, | \varphi(x) \rangle = \cos(2\pi \Lambda_{\bar{j}} x)$.

It is possible to express $O((N')^{L-1})$ different frequencies in this way, assuming the $\Lambda_{\bar{j}}$ are distinct, which will be the case for example with probability 1 if the $\{\lambda_{\ell i}\}$ are drawn i.i.d. from some distribution with continuous support. This further motivates the small $L$ regime where exponential advantage in communication is possible. These types of circuits with interleaved data-dependent unitaries and parameterized unitaries was considered for example in [104], and is also related to the setting of quantum signal processing and related algorithms [74, 77]. We also show that such circuits can express dense function in Fourier space, and for small $N$ we additionally find that these circuits are universal function approximators (Appendix F.2), though in this setting the possible communication advantage is less clear.

The problem of applying nonlinearities to data encoded efficiently in quantum states is non-trivial and is of interest due to the importance of nonlinearities in enabling efficient function approximation [76]. One approach to resolving the constraints of unitarity with the potential irreversibility of nonlinear functions is the introduction of slack variables via additional ancilla qubits, as typified by the techniques of block-encoding [29, 41]. Indeed, these techniques can be used to apply nonlinearities to amplitude encoded data efficiently, as was recently shown in [100]. This approach can be applied to the distributed setting as well. Consider the communication problem where Alice is given $x$ as input and Bob is given unitaries $\{U_1, U_2\}$ over $\log N$ qubits. Denote by $\sigma : \mathbb{R} \to \mathbb{R}$ a nonlinear function such as the sigmoid, exponential or standard trigonometric functions, and $n = 2^N$. We show the following:

**Lemma 10.** *There exists a model $|\varphi_\sigma\rangle$ of the form definition 3.1 with $L = O(\log 1/\varepsilon), N' = 2^{n'}$ where $n' = 2n + 4$ such that $|\varphi_\sigma\rangle = \alpha \, |0\rangle^{\otimes n+4} \, |\hat{y}\rangle + |\phi\rangle$ for some $\alpha = O(1)$, where $|\hat{y}\rangle$ is a state that obeys*

$$\left\| |\hat{y}\rangle - \left| U_2 \frac{1}{\|\sigma(U_1 x)\|_2} \sigma(U_1 x) \right\rangle \right\|_2 < \varepsilon. \tag{F.3}$$

$|\phi\rangle$ *is a state whose first $n + 4$ registers are orthogonal to $|0\rangle^{\otimes n+4}$.*

Proof: Appendix B.

This result implies that with constant probability, after measurement of the first $n + 4$ qubits of $|\varphi_\sigma\rangle$, one obtains a state whose amplitudes encode the output of a single hidden layer neural network. It may also be possible to generalize this algorithm and apply it recursively to obtain a state representing a deep feed-forward network with unitary weight matrices.

It is also worth noting that the general form of the circuits we consider resembles self-attention based models with their nonlinearities removed (motivated for example by [107]), as we explain in Appendix F.3. Finally, in Appendix F.4 we discuss other strategies for increasing the expressivity of these quantum circuits by combining them with classical networks.

## F.2 Additional results on oscillatory features

Extending the unitaries considered in Appendix F.1 to more than one variable, for two scalar variables $x, y$ define

$$A_\ell(x) \quad = \text{diag}((e^{-2\pi i \lambda_{\ell 1} x}, \ldots, e^{-2\pi i \lambda_{\ell N'} x})), \tag{F.4a}$$

$$A_\ell(x, y) \quad = \text{diag}((e^{-2\pi i \lambda_{\ell 1} x}, \ldots, e^{-2\pi i \lambda_{\ell, N'/2} x}, e^{-2\pi i \lambda_{\ell, N'/2+1} y}, \ldots, e^{-2\pi i \lambda_{\ell N'} y})) \tag{F.4b}$$

for $\ell \in \{1, \ldots, L\}$. Once again $\{B_\ell\}$ are data-independent unitaries, and we denote by $|\varphi(x)\rangle, |\varphi(x, y)\rangle$ the states defined by interleaving these unitaries in the manner of eq. (3.1), and by $\mathcal{L}_1, \mathcal{L}_2$ the corresponding loss functions when measuring $X_0$.

While the circuits in Appendix F.1 enable one to represent a small number of frequencies from a set that is exponential in $L$, one can easily construct circuits that are supported on an exponentially large number of frequencies, as detailed in Lemma 11. We also use measures of expressivity of classical neural networks known as *separation rank* to show that circuits within the class eq. (3.1) can represent complex correlations between their inputs. For a function $f$ of two variables $y, z$, its

separation rank is defined by

$$\text{sep}(f) \equiv \min \left\{ R : f(x) = \sum_{i=1}^{R} g_i(y)h_i(z) \right\}. \tag{F.5}$$

If for example $f$ cannot represent any correlations between $y$ and $z$, then $\text{sep}(f) = 1$. When computed for certain classes of neural networks, $y, z$ are taken to be subsets of a high-dimensional input. The separation rank can be used for example to quantify the inductive bias of convolutional networks towards learning local correlations [33], the effect of depth in recurrent networks [69], and the ability of transformers to capture correlations across sequences as a function of their depth and width [70].

We find that the output of estimating an observable using circuits of the form eq. (F.4) can be supported on an exponential number of frequencies, and consequently has a large separation rank:

**Lemma 11.** *For $\{\lambda_{\ell i}\}$ drawn i.i.d. from any continuous distribution and parameterized unitaries $\{B_\ell\}$ such that the real and imaginary parts of each entry in these matrices is a real analytic function of parameters $\Theta$ drawn from a subset of $\mathbb{R}^{PL}$, aside from a set of measure $0$ over the choice of $\{\lambda_{\ell i}\}, \{B_\ell\}$,*

*i) The number of nonzero Fourier components in $\mathcal{L}_1$ is $\left( \frac{N'(N'-1)}{2} \right)^{L-1} N'$.*

*ii)*

$$\text{sep}(\mathcal{L}_2) = 2 \left( \frac{N'(N'-1)}{2} \right)^{L-1} N'. \tag{F.6}$$

Proof: Appendix B

This almost saturates the upper bound on the number of frequencies that can be expressed by a circuit of this form that is given in [104]. The separation rank implies that complex correlations between different parts of the sequence can in principle be represented by such circuits. The constraint on $\{B_\ell\}$ is quite mild, and applies to standard choices of parameterize unitaries.

The main shortcoming of a result such as Lemma 11 is that it is not robust to measurement error as it is based on constructing states that are equal weight superpositions of an exponential number of terms. It is straightforward to show that circuits of this form can serve as universal function approximators, at least for a small number of variables. For high-dimensional functions it is unclear when a communication advantage is possible, as we describe below.

**Lemma 12.** *Let $f$ be a $p$-times continuously differentiable function with period $1$, and denote by $\hat{f}_{:M}$ the vector of the first $M$ Fourier components of $f$. If $\left\| \hat{f}_{:M} \right\|_1 = 1$ then there exists a circuit of the form eq. (3.1) over $O(\log M)$ qubits such that*

$$\|\mathcal{L} - f\|_\infty \leq \frac{C}{M^{p-1/2}} \tag{F.7}$$

*for some absolute constant $C$.*

Proof: Appendix B

This result improves upon the result in [93, 104] about universal approximation with similarly structured circuits both because it is non-asymptotic and because it shows uniform convergence rather than convergence in $L_2$. Non-asymptotic results universal approximation results were also obtained recently by [43], however their approximation error scales polynomially with the number of qubits, as opposed to exponentially as in Lemma 12.

The result of Lemma 12 applies to an $L = 1$ circuit. The special hierarchical structure of the Fourier transform implies that the same result can be obtained using even simpler circuits with larger $L$. Consider instead single-qubit data-dependent unitaries over $L + 1$ qubits that take the form

$$A_\ell = |0\rangle_0 \langle 0|_0 + |1\rangle_0 \langle 1|_0 \left( |0\rangle_{\ell+1} \langle 0|_{\ell+1} + e^{2\pi i 2^{\ell-1} x} |1\rangle_{\ell+1} \langle 1|_{\ell+1} \right), \tag{F.8}$$

for $\ell \in \{1, \ldots, L\}$. This is simply a single term in a hierarchical decomposition of the same feature matrix we had in the shallow case, since

$$\prod_{\ell=1}^{L} A_\ell = |0\rangle_0 \langle 0|_0 \otimes I_{1:L} + |1\rangle_0 \langle 1|_0 \otimes I_1 \otimes \left( \sum_{m=0}^{2^L-1} e^{2\pi i m x} |m\rangle \langle m| \right), \tag{F.9}$$

which is identical to eq. (B.45). As before, set

$$B_1 = |\hat{f}\rangle \langle 0| + |0\rangle \langle \hat{f}|, \tag{F.10}$$

with $N'/4 = 2^L$ and $B_\ell = I$ for $\ell > 1$. This again gives an approximation of $f$ up to normalization. The data-dependent unitaries are particularly simple when decomposed in this way. The fact that they act on a single qubit and thus have "small width" is reminiscent of classical depth-separation result such as [32], where it is shown that (roughly speaking) within certain classes of neural network, in order to represent the function implemented by a network of depth $L$, a shallow network must have width exponential in $L$. In this setting as well the expressive power as measured by the convergence rate of the approximation error grows exponentially with $L$ by eq. (B.44).

The circuits above can be generalized in a straightforward way to multivariate functions of the form $f : [-1/2, 1/2]^D \to \mathbb{R}$ and combined with multivariate generalization of eq. (B.44). In this case the scalar $m$ is replaced by a $D$-dimensional vector taking $M^D$ possible values, and we can define

$$A_1(x) = |0\rangle_0 \langle 0|_0 \otimes I_{1:D \log M - 1} + |1\rangle_0 \langle 1|_0 \otimes I_1 \otimes \left( \sum_{m \in [M]^D} e^{2\pi i m \cdot x} |m\rangle \langle m| \right). \tag{F.11}$$

Note that using this feature map, the number of neurons is linear in the spatial dimension $D$. Because of this, such circuits are not strictly of the form eq. (3.1) for general $N$ since it is *not* the case that $\log N' = O(\log N)$ where $N'$ is the Hilbert space on which the unitaries in the circuit act and $N$ is the size of $x$. An alternative setting where the features themselves are also learned from data could enable much more efficient approximation of functions that are sparse in Fourier space.

### F.3 Unitary Transformers

Transformers based on self-attention [109] form the backbone of large language models [25, 16] and foundation models more generally [20]. A self-attention layer, which is the central component of transformers, is a map between sequences in $\mathbb{R}^{S \times N'}$ (where $S$ is the sequence length) defined in terms of weight matrices $W_Q, W_K, W_V \in \mathbb{R}^{N' \times N'}$, given by

$$X'(X) = \text{softmax}\left( \frac{X W_Q W_K^T X^T}{\sqrt{N}} \right) X W_V \equiv A(X) X W_V, \tag{F.12}$$

where $\text{softmax}(x)_i = e^{x_i} / \sum_i e^{x_i}$ for a vector $x$, and acts row-wise on matrices. There is an extensive literature on replacing the softmax-based attention matrix $A(X)$ with matrices that can be computed more efficiently, which can markedly improve the time complexity of inference and training without a significant effect on performance [63, 70]. In some cases $A(X)$ is replaced by a unitary matrix [68]. Remarkably, recent work shows that models without softmax layers can in fact outperform standard transformers on benchmark tasks while enabling faster inference and a reduced memory footprint [107].

Considering a simplified model that does not contain the softmax operation as in [70] and dropping normalization factors, the linear attention map is given by

$$X'_{\text{lin}}(X) = X W_Q W_K^T X^T X W_V. \tag{F.13}$$

Iterating this map twice gives

$$X'_{\text{lin}}(X'_{\text{lin}}(X)) = \begin{array}{l} X W_Q^{(1)} W_K^{(1)T} X^T X W_V^{(1)} W_Q^{(2)} W_K^{(2)T} W_V^{(1)T} X^T X * \\ W_K^{(1)} W_Q^{(1)T} X^T X W_Q^{(1)} W_K^{(1)T} X^T X W_V^{(1)} W_V^{(2)}. \end{array} \tag{F.14}$$

Iterating this map $K$ times (with different weight matrices at each layer) gives a function of the form:

$$X_{\text{lin}}^{(K)}(X) = X R_0 \prod_{\ell=1}^{(3^K-1)/2} \left( X^T X R_\ell \right), \tag{F.15}$$

where the $\{R_\ell\}$ matrices depend only on the trainable parameters. If we now constrain these to be parameterized unitary matrices, and additionally replace $X^T X$ with a unitary matrix $U_X$ encoding features of the input sequence itself, then the $i$-th row of the output of this model is encoded in the amplitudes of a state of the form eq. (3.5) with $L = (3^K - 1)/2 + 1, |\psi(x)\rangle = |X_i\rangle, A_\ell(x) = U_X, B_\ell = R_\ell$.

### F.4 Ensembling and point-wise nonlinearities

An additional method for increasing expressivity while maintaining an advantage in communication is through ensembling. Given $K$ models of the form Definition 3.1 with $P$ parameters each, one can combine their loss functions $\mathcal{L}_1, \ldots, \mathcal{L}_K$ into any differentiable nonlinear function

$$\tilde{\mathcal{L}}(\mathcal{L}_1(\Theta_1, x), \ldots, \mathcal{L}_K(\Theta_K, x), \tilde{\Theta}, x) \tag{F.16}$$

that depends on additional parameters $\tilde{\Theta}$. As long as $K$ and $|\tilde{\Theta}|$ scale subpolynomially with $N$ and $P$, the gradients for this more expressive model can be computed while maintaining the exponential communication advantage in terms of $N, P$.

## G    Realizing quantum communication

Given the formidable engineering challenges in building a large, fault tolerant quantum processor [11, 44], the problem of exchanging coherent quantum states between such processors might seem even more ambitious. We briefly outline the main problems that need to be solved in order to realize quantum communication and the state of progress in this area, suggesting that this may not be the case.

We first note that sending a quantum state between two processors can be achieved by the well-known protocol of quantum state teleportation [18, 45]. Given an $n$ qubit state $|\psi\rangle$, Alice can send $|\psi\rangle$ to Bob by first sharing $n$ Bell pairs of the form

$$|b\rangle = \frac{1}{\sqrt{2}} \left( |0\rangle |0\rangle + |1\rangle |1\rangle \right), \tag{G.1}$$

(sharing such a state involves sending a one of the two qubits to Bob) and subsequently performing local processing on the Bell pairs and exchanging $n$ bits of classical communication. Thus quantum communication can be reduced to communicating Bell pairs up to a logarithmic overhead, and does not require say transmitting an arbitrary quantum state in a fault tolerant manner, which appears to be a daunting challenge given the difficulty of realizing quantum memory on a single processor. Bell pairs can be distributed by a third party using one-way communication.

In order to perform quantum teleportation, the Bell pairs must have high fidelity. As long as the fidelity of the communicated Bell pairs is above .5, purification can be used produce high fidelity Bell pairs [19], with the fidelity of the purified Bell pair increasing exponentially with the number of pairs used. Thus communicating arbitrary quantum states can be reduced to communicating noisy Bell pairs.

Bell pair distribution has been demonstrated across multiple hardware platforms including superconducting waveguides [75], optical fibers [64], free space optics at distances of over $1, 200$ kilometers [71]. At least in theory, even greater distances can be covered by using quantum repeaters, which span the distance between two network nodes. Distributing a Bell pair between the nodes can then be reduced to sharing Bell pairs only between adjacent repeaters and local processing [12].

A major challenge in implementing a quantum network is converting entangled states realized in terms of photons used for communication to states used for computation and vice versa, known as transduction [66]. Transduction is a difficult problem due to the several orders of magnitude in energy that can separate optical photons from the energy scale of the platform used for computation. Proof of principle experiments have been performed across a number of platforms including trapped ions [64], solid-state systems [95], and superconducting qubits operating at microwave frequencies [14, 112].

## H    Privacy of Quantum Communication

In addition to an advantage in communication complexity, the quantum algorithms may potentially lead to advantages in terms of privacy. It is well known that the number of bits of information that

can be extracted from an unknown quantum state is proportional to the number of qubits. It follows immediately that since the above algorithm requires exchanging a logarithmic number of copies of states over $O(\log N)$ qubits, even if all the communication between the two players is intercepted, an attacker cannot extract more than a logarithmic number of bits of classical information about the input data or model parameters. Specifically, we have:

**Corollary 1.** *If Alice and Bob are implementing the quantum algorithm for gradient estimation described in Lemma 2, and all the communication between Alice and Bob is intercepted by an attacker, the attacker cannot extract more than $\tilde{O}(L^2(\log N)^2(\log P)^2 \log(L/\delta)/\varepsilon^4)$ bits of classical information about the inputs to the players.*

This follows directly from Holevo's theorem [52], since the multiple copies exchanged in each round of the protocol can be thought of as a quantum state over $\tilde{O}((\log N)^2(\log P)^2 \log(L/\delta)/\varepsilon^4)$ qubits. As noted in [3], this does not contradict the fact that the protocol allows one to estimate all $P$ elements of the gradient, since if one were to place some distribution over the inputs, the induced distribution over the gradient elements will generally exhibit strong correlations. An analogous result holds for the inference problem described in Lemma 1.

It is also interesting to ask how much information either Bob or Alice can extract about the inputs of the other player by running the protocol. If this amount is logarithmic as well, it provides additional privacy to both the model owner and the data owner. It allows two actors who do not necessarily trust each other, or the channel through which they communicate, to cooperate in jointly training a distributed model or using one for inference while only exposing a vanishing fraction of the information they hold.

It is also worth mentioning that data privacy is also guaranteed in a scenario where the user holding the data also specifies the processing done on the data. In this setting, Alice holds both data $x$ and a full description of the unitaries she wishes to apply to her state. She can send Bob a classical description of these unitaries, and as long as the data and features are communicated in the form of quantum states, only a logarithmic amount of information can be extracted about them. In this setting there is of course no advantage in communication complexity, since the classical description of the unitary will scale like $\text{poly}(N, P)$.

# I   Some Open Questions

## I.1   Expressivity

Circuits that interleave parameterized unitaries with unitaries that encode features of input data are also used in Quantum Signal Processing [74, 77], where the data-dependent unitaries are time evolution operators with respect to some Hamiltonian of interest. The original QSP algorithm involved a single parameterized rotation at each layer, and it is also known that extending the parameter space from $U(1)$ to $SU(2)$ by including an additional rotation improves the complexity of the algorithm and improves its expressivity [85]. In both cases however the expressive power (in terms of the degree of the polynomial of the singular values that can be expressed) grows only linearly with the number of interleaved unitaries. Given the natural connection to the distributed learning problems considered here, it is interesting to understand the expressive power of such circuits with more powerful multi-qubit parameterized unitaries.

We present a method of applying a single nonlinearity to a distributed circuit using the results of [100]. Since this algorithm requires a state-preparation unitary as input and produces a state with a nonlinearity applied to the amplitudes, it is natural to ask whether it can be applied recursively to produce a state with the output of a deep network with nonlinearties encoded in its amplitudes. This will require extending the results of [100] to handle noisy state-preparation unitaries, yet the effect of errors on compositions of block encodings [29, 41], upon which these results are based, is relatively well understood. It is also worth noting that these approaches rely on the approximation of nonlinear functions by polynomials, and so it may also be useful to take inspiration directly from classical neural network polynomial activations, which in some settings are known to outperform other types of nonlinearities [80].

## I.2 Optimization

The results of Appendix D rely on sublinear convergence rates for general stochastic optimization of convex functions (Lemma 8). It is known however that using additional structure, stochastic gradients can be used to obtain linear convergence (meaning that the error decays exponentially with the number of iterations). This is achievable when subsampling is the source of stochasticity [67], or with occasional access to noiseless gradients in order to implement a variance reduction scheme [60, 84, 46], neither of which seem applicable to the setting at hand. It is an interesting open question to ascertain whether there is a way to exploit the structure of quantum circuits to obtain linear convergence rates using novel algorithms. Aside from advantages in time complexity, this could imply an exponential advantage in communication for a more general class of circuits.

Conversely, it is also known that given only black-box access to a noisy gradient oracle, an information-theoretic lower bound of $\Omega(1/T)$ on the error holds given $T$ oracle queries, precluding linear convergence without additional structure, even for strongly convex objectives [9]. [49] provide a similar lower bound for their algorithm, at least for a restricted class of circuits. Perhaps these results be used to show optimality of algorithms that rely on the standard variational circuit optimization paradigm that involves making quantum measurements at every iteration and using these to update the parameters. This might imply that linear convergence is only possible if the entire optimization process is performed coherently.

In this context, we note that the treatment of gradient estimation at every layer and every iteration as an independent shadow tomography problem is likely highly suboptimal, since no use is made of the correlations across iterations between the states and the observables of interest. While in Appendix D.2 this is not the case, that algorithm applies only to fine-tuning of a single layer. Is there a way to re-use information between iterations to reduce the resource requirements of gradient descent using shadow tomography? One approach could be warm-starting the classical resource states by reusing them between iterations. Improvements along these lines might find applications for other problems as well.

## I.3 Exponential advantage under weaker assumptions

The lower bound in Lemma 3 applies to circuits that contain general unitaries, and thus have depth $\text{poly}(N)$ when compiled using a reasonable gate set. One can ask whether the lower bound can be strengthened to apply to more restricted classes of unitaries as well, and in particular $\log$-depth unitaries. While it is known that exponential communication advantages require the circuits to have $\text{poly}(N)$ gate complexity overall [7], this does not rule out the possibility of computational separations resulting from the clever encoding and transmission of states nor does it rule out communication advantages resulting from very short time preparations from $\log$-depth protocols. The rapid growth of complexity of random circuits composed from local gates with depth suggests that this might be possible [24]. This is particularly interesting since Algorithm 1 requires only a single measurement per iteration and may thus be suitable for implementation on near-term devices whose coherence times restrict them to implementing shallow circuits. It has also been recently shown that an exponential quantum advantage in communication holds for a problem which is essentially equivalent to estimating the loss of a circuit of the form Definition 3.1 with $L = 2$, under a weaker model of quantum communication than the standard one we consider [10]. This is the one-clean-qubit model, in which the initial state $|\psi(x)\rangle$ consists of a single qubit in a pure state, while all other qubits are in a maximally mixed state.

# J Experiments additional details

## J.1 Node classification training

We use the same training regime for all datasets using the recommended hyperparameters in DGL [113] examples, reported in Table 3.

We trained each model 10 times for all three datasets using a single NVIDIA RTX A6000, taking approximately 15 minutes per execution.

Table 3: Hyper parameters of node classification training

| Hyperparameter | Value |
|---|---|
| Hidden dimension | 512 |
| SIGN hops | 5 |
| Learning rate | 0.001 |
| Input dropout | 0.3 |
| Hidden dropout | 0.4 |
| Weight decay | 0.0 |

Table 4: Hyper parameters of node classification training

| Hyperparameter | Values |
|---|---|
| Hidden dimension | 8,12,16,32,64,96,128,148,256 |
| SIGN hops (per operation) | [0-10] |
| Learning rate | 0.001, 0.003, 0.005, 0.01, 0.03, 0.05, 0.1 |
| Input dropout | 0.0 |
| Hidden dropout | 0.0, 0.1, 0.2, 0.3, 0.4, 0.5 |
| Weight decay | 0.0, 1e-8, 1e-7, 1e-6, 1e-5, 1e-4 |
| Batch size | 32, 64, 128, 256, 512, 1024 |
| Normalization layer | BatchNorm, LayerNorm, none |

## J.2 Graph classification training

As, to the best of our knowledge, we are the first to use a SIGN variant on graph classification tasks, we conducted a comprehensive hyperparameter tuning for the model structure (including the number of message passing operators, the hidden dimension, and normalization after the hidden layer) and optimization settings. The tuning was performed using Bayesian hyperparameter optimization to identify the optimal values for each dataset. This process involved varying the hidden dimension, the number of SIGN hops per operation, the learning rate, and dropout rates. The values considered for each hyperparameter are detailed in Table 4. The full results of these experiments are in Table 5.

We scan each task for approximately 150 runs, using a single NVIDIA RTX A6000.

Table 5: Graph Classification Test Accuracy. Our model achieves comparable results to GIN and other known models on most datasets.

| | Dataset | | | | | | | | |
|---|---|---|---|---|---|---|---|---|---|
| Model | MUTAG | PTC | NCI1 | PROTEINS | COLLAB | IMDB-B | IMDB-M | REDDIT-B | REDDIT-M |
| GIN [114] | 89.40±5.60 | 64.60±7.0 | 82.17±1.7 | 76.2 ±2.8 | 80.2 ±1.90 | 75.1 ±5.1 | 52.3 ±2.8 | 92.4 ±2.5 | 57.5±1.5 |
| DropGIN[92] | 90.4 ±7.0 | 66.3 ±8.6 | - | 76.3 ±6.1 | - | 75.7 ±4.2 | 51.4 ±2.8 | - | - |
| DGCNN[118] | 85.8 ±1.7 | 58.6 ±2.5 | - | 75.5 ±0.9 | - | 70.0 ±0.9 | 47.8 ±0.9 | - | - |
| U2GNN [89] | 89.97±3.65 | 69.63±3.60 | - | 78.53±4.07 | 77.84±1.48 | 77.04±3.45 | 53.60±3.53 | - | - |
| HGP-SL[119] | - | - | 78.45±0.77 | 84.91±1.62 | - | - | - | - | - |
| WKPI[120] | 88.30±2.6 | 68.10±2.4 | 87.5 ±0.5 | 78.5±0.4 | - | 75.1 ±1.1 | 49.5 ± 0.4 | - | 59.5 ± 0.6 |
| SIGN (ours) | 92.02±6.45 | 68.0 ±8.17 | 77.25±1.42 | 76.55±5.10 | 81.82±1.42 | 76 ±2.49 | 53.13±3.01 | 78.95±2.72 | 54.09±1.76 |

## J.3 Empirical bounds

We measure $\|W_1\|$ and $\|W_2\|_\infty$ of the trained graph classification models in Section 5.2.1, corresponding to Equation (4.2) and report the average results over 10 runs in Table 6 (note that we use $P = I$ so that no pooling matrix is present, and in any case the pooling window will typically be a small constant). $W_1$ is constructed as a block diagonal matrix of the weights of the SIGN hidden layer. $\|W_2\|_\infty$ is the infinity norm of the weight matrix of the output layer of SIGN, multiplied by 2 (since we compute differnces between numbers of nodes in two classes).

We measure the score difference of the graph classification task in Section 5.2.1 and compare them to the differences of the class sizes in Figure 3. Most of the differences are significant (larger than $1/\text{poly}(N)$ where $N$ is the number of nodes; see fig. 3(c)). Some class pairs have low differences making them indistinguishable, however, fig. 3(a),(b) indicate those are typically classes with similar

Table 6: Weight norms of the graph classification models. We measure the norms of the final decision problem models, averaging the values over 8 runs.

| | Dataset | | |
|---|---|---|---|
| Value | OGBN-Products | Reddit | Cora |
| $\|W_1\|$ | $3.46 \pm 0.27$ | $1.32 \pm 0.12$ | $8.5 \pm 8.0$ |
| $\|W_2\|_\infty$ | $0.13 \pm 0.02$ | $0.04 \pm 0.00$ | $0.1 \pm 0.07$ |
| #nodes | 2,449,029 | 232,965 | 2708 |

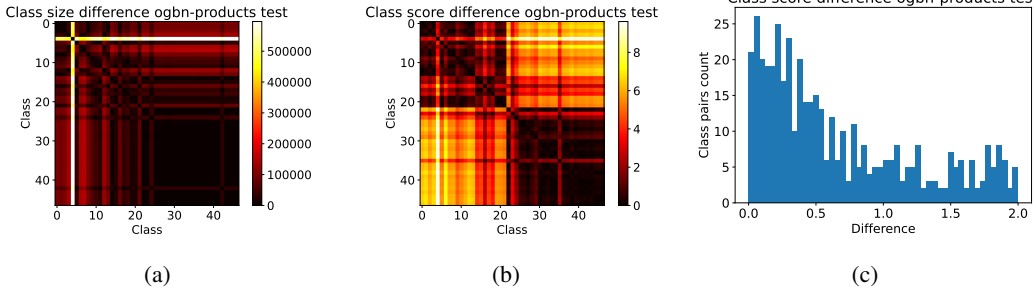

(a)                         (b)                         (c)

Figure 3: (a): Difference between class sizes in ogbn-products test set. (b) Difference between the graph classification model class scores. The score differences are correlated to the class size differences. (c) Histogram of the class pairs differences. Most of the differences are significantly larger than $1/N$.

number of nodes. This provides evidence that when there is a considerable class imbalance (i.e. one that scales with system size), the magnitude of the model output when computing this difference will not decay with the size of the graph.

While evidence of asymptotic scaling will require experiments on graphs of different sizes, our results suggest that the upper bound in Lemma 6 is not large for models trained on standard benchmarks, implying that they can be efficiently simulated on a quantum computer.

