# OpenReview forum: "Exponential Quantum Communication Advantage in Distributed Inference and Learning"
_NeurIPS.cc/2024/Conference — NeurIPS 2024 poster_

### Official Review · Reviewer_K89H · 2024-07-11

**Soundness:** 4
**Presentation:** 4
**Contribution:** 2
**Rating:** 6
**Confidence:** 4

**Summary:**

This paper studies distributed learning and inference based on graph neural networks but with quantum communications between the distributed agents. The authors show that quantum networks reduce the communication complexity inference and gradient computation of distributed models. This reduction is shown to be exponential in communication complexity. The main idea of the paper is to have the data encoded into qubits using amplitude encoding. With that one gets an exponential reduction in space and hence communication complexity. As for the gradient estimation, the problem is shown to be converted to Aaronson's Shadow tomography.

**Strengths:**

The paper proposes an exponential advantage in communication complexity for the task of distributed inference and gradient estimation. This is a nice set of advantages that can be employed in practical problems. Also the lower bounds are nice contributions.

**Weaknesses:**

The result of the paper is not surprising given the known results in quantum communication complexity. Moreover, the results seem to be directly derived from known papers on Shadow tomography and the Threshold search.

**Questions:**

Can you elaborate on the run-time of the proposed method?

**Limitations:**

Nothing significant.

---

> ### Author Rebuttal · Authors · 2024-08-05
>
> > Can you elaborate on the run-time of the proposed method?
>
>  The run-time of gradient estimation based on shadow tomography is stated in Theorem 2. Asymptotically, this will dominate the cost of other parts of the algorithms such as state preparation. This polynomial scaling is based on the complexity of solving SDPs, based on the paper of Brandao et al (reference 22 in the submission). This algorithm is used as part of the online state learning procedure of shadow tomography. This scaling can be improved in cases where the unitaries have low rank structure. The coordinate descent based algorithm we consider in Algorithm 1 will enjoy better scaling in certain parameter regimes (namely when the vector of coefficients of the unitary parameters has small norm).

---

> > ### Comment · Reviewer_K89H · 2024-08-11
> >
> > Thank you for your reply. I will keep my score.

---

### Official Review · Reviewer_mkaP · 2024-07-15

**Soundness:** 3
**Presentation:** 3
**Contribution:** 3
**Rating:** 7
**Confidence:** 4

**Summary:**

The paper investigates the exponential quantum advantage in communication complexity over the tasks of training or inference in multiple types of compositional distributed quantum circuit model. The authors also study the quantum communication advantage in a specific class of shallow graph neural networks, and then they conduct experiments showing that it can achieve performance comparable to certain existing classical models and benchmarks while maintaining the exponential communication advantage. At the end, the authors explore the expressivity of the compositional model.

**Strengths:**

I think this paper is one of the good examples of applying results and tools from quantum complexity theory to problems in distributed computing.

**Weaknesses:**

(1) Lemma 6 addresses the quantum communication complexity of the quadratic graph network inference problem, for which I think a lower bound would make more sense. However, (I am not sure if I miss anything or not), instead of giving a lower bound, Lemma 6 provides an upper bound, which I find confusing.
(2) The paper assumes that the input is amplitude encoded. It is well-known that preparing such quantum states is generally not efficient. This is an important factor to consider regarding the feasibility of the proposed systems, and thus, it is my main concern with the paper. However, I don't see much formal treatment of this issue in the paper.

**Questions:**

My questions are those 2 points in Weaknesses.

**Limitations:**

Yes, the authos give a fair discussion on possible limitations.

---

> ### Author Rebuttal · Authors · 2024-08-05
>
> (1) We consider the problem of inference with graph networks in order to demonstrate the the communication advantage holds for model classes that can be trained on classical data. Note that in order to show the advantage, we must show both that any classical algorithm will require lots of communication (Lemma 5) and that quantum algorithms exist that require little communication. The latter is the content of Lemma 6, hence it takes the form of an upper bound (as opposed to the classical lower bound).
>
> (2) Indeed, the paper requires loading the data onto a quantum computer. Note that the time complexity is only polynomial in the Hilbert space dimension, as mentioned in Theorem 2. While this is typically considered inefficient for quantum algorithms (since it is exponential in the number of qubits), the cost of classical inference and training is in fact polynomial in the data and model size, so this scaling is only polynomially worse than the classical scaling. The coordinate descent algorithm we consider in Algorithm 1 is more efficient than this. In general the cost of loading an input state will be linear in N. Note that while such scaling can eliminate any advantages in time complexity, which is a major issue for many proposals of quantum advantages related to machine learning problems, it does not affect the communication complexity, in terms of which exponential advantage is still possibile.

---

> > ### Comment · Reviewer_mkaP · 2024-08-14
> >
> > Thanks for the response. As for comments on W2, do you mean in general performing amplitude encoding of input data takes time linear in N?

---

### Official Review · Reviewer_W5VQ · 2024-07-23

**Soundness:** 3
**Presentation:** 3
**Contribution:** 2
**Rating:** 6
**Confidence:** 2

**Summary:**

This paper studies the quantum advantage on communication of distributed learning. It places a common quantum neural network model in the (two-party) communication scenario, where the data x is assigned to Alice, and the parameterized unitary operators of each layer are alternately given to Alice and Bob in order.

The paper shows exponential quantum advantage on communication for problems of estimating the loss function (some function on data after the action of unitary operators) and estimating the gradients of the loss function with respect to each parameter of the unitary operators. The quantum upper bound for estimating the loss function is by encoding data x (size-N vector) into O(log N)-qubit state. The quantum upper bound for estimating the gradients is by a reduction to a shadow tomography problem. The classical lower bound is by a reduction from a problem proposed by Raz, also a round-based lower bound by a reduction from pointer-chasing.

For comparison, the paper shows that no exponential quantum advantage on communication of linear classification, where Alice and Bob are given data x and y as vectors to determine the sign of inner product of x and y. The classical upper bound is by applying Johnson-Lindenstrauss and the quantum lower bound is by a reduction from gap-Hamming.

The paper also shows exponential quantum advantage in communcication for quadratic graph network inference, and provides experimental results for the above scenarios.

**Strengths:**

- A new research direction on quantum advantage of communication complexity for learning problems, providing a series of results with solid theoretical proofs and experimental data to support them.

**Weaknesses:**

- For technical part, the upper bounds are straightforward, and the lower bound reductions look standard.
- These learning tasks (estimating the loss function in quantum neural networks, linear classification, and quadratic graph network inference) seem to have weak correlations, making the paper appear somewhat like a mere compilation of results.

**Questions:**

- Minor comment: the upper bound in Lemma 1 should be O(L log N) instead of O(log N)?

**Limitations:**

- The claim of QUANTUM advantage on QUANTUM neural networks seems a bit like cheating, although the model and problem are described using classical information.
- The two-party version of quantum neural networks seems artificial to me. The data and parameters of each single unitary operator might be stored distributively between parties, just like in problem 7 of linear classification, where data x and y are assigned to Alice and Bob.

---

> ### Author Rebuttal · Authors · 2024-08-05
>
> > the upper bound in Lemma 1 should be O(L log N) instead of O(log N)?
>
> The total communication is indeed linear in L, but for this reason the number of rounds required is linear in L. In each round, log(N) qubits should suffice in order to solve the problem.
>
> > The claim of QUANTUM advantage on QUANTUM neural networks seems a bit like cheating, although the model and problem are described using classical information.
>
> The problems we consider involve classical data, which can be encoded in the amplitudes of quantum states. Similarly, classical neural networks can be expressed as quantum circuits. Our results indicate that for certain model classes, doing this can lead to exponential savings in communication for distributed training. The overarching motivation is indeed the solution of classical learning problems. Note that in order to enjoy these advantages, we require that the quantum algorithm simply perform equivalently to the classical algorithm in terms of the accuracy or the time complexity, and we show that for such models exponential communication advantages are still possible. If we have misunderstood the concern that is being expressed here, we will be happy to elaborate further as needed during the discussion phase.
>
> > The two-party version of quantum neural networks seems artificial to me. The data and parameters of each single unitary operator might be stored distributively between parties, just like in problem 7 of linear classification, where data x and y are assigned to Alice and Bob.
>
> The motivation for the model is the structure of distributed classical neural networks. Note that pipeline parallelism is standard in training of large models, and is particularly useful in cases where the hardware is heterogeneous and one wants to sparsely activate subsets of a large model. We believe our distributed quantum model is the natural quantum analog of this type of neural network. Indeed the expressivity of such models is unclear, which is why we additionally consider a specific class of graph neural networks in Section 4, for which the communication advantage holds.

---

> > ### Comment · Reviewer_W5VQ · 2024-08-09
> >
> > Thank you for the response. Indeed the classical neural networks can be expressed as quantum circuits, but I am not sure whether converting classical neural networks into quantum circuits introduces much additional communication in your distributed setting.

---

> > > ### Author Response · Authors · 2024-08-09
> > >
> > > In our setting, each party constructs a set of unitaries locally, requiring no communication. Note that each unitary can be arbitrarily complex, since this is done locally. Each unitary can also be a nonlinear function of classical data or model parameters. The total communication complexity of inference will then scale linearly with the number of times the quantum state encoding the features has to be transferred back and forth between the players when evaluating the resulting circuit.
> > >
> > > A possible concern here is that it might be hard to construct expressive models in this way, or models that approximate the action of realistic classical neural networks. We show in section 4 however that for certain classes of graph GNNs, models of this form require only a single round of communication to run inference and are expressive enough to perform well on standard benchmarks.

---

### Author Rebuttal · Authors · 2024-08-05

We thank the reviewers for their helpful comments and endorsement of the work. We will respond to individual reviewers below.

---

### Decision · Program_Chairs · 2024-09-25

**Decision:**

Accept (poster)

**Comment:**

This paper studied distributed inference and learning in the setting of quantum computing. In particular, exponential quantum communication advantage is proved, and experiments also supported the theoretical claims. The review scores are unianimously positive and the decision is to accept this paper at NeurIPS 2024. For the final version, the authors should carefully incorporate all the points raised during rebuttal for better clarification.